# HSF1 phosphorylation establishes an active chromatin state via the TRRAP–TIP60 complex and promotes tumorigenesis

Mitsuaki Fujimoto [1], Ryosuke Takii [1], Masaki Matsumoto [2], Mariko Okada[1], Keiich I. Nakayama [3], Ryuichiro Nakato [4], Katsunori Fujiki[4], Katsuhiko Shirahige[4] & Akira Nakai [1] ✉

Transcriptional regulation by RNA polymerase II is associated with changes in chromatin structure. Activated and promoter-bound heat shock transcription factor 1 (HSF1) recruits transcriptional co-activators, including histone-modifying enzymes; however, the mechanisms underlying chromatin opening remain unclear. Here, we demonstrate that HSF1 recruits the TRRAP-TIP60 acetyltransferase complex in *HSP72* promoter during heat shock in a manner dependent on phosphorylation of HSF1-S419. TRIM33, a bromodomain-containing ubiquitin ligase, is then recruited to the promoter by interactions with HSF1 and a TIP60-mediated acetylation mark, and cooperates with the related factor TRIM24 for mono-ubiquitination of histone H2B on K120. These changes in histone modifications are triggered by phosphorylation of HSF1-S419 via PLK1, and stabilize the HSF1-transcription complex in *HSP72* promoter. Furthermore, HSF1-S419 phosphorylation is constitutively enhanced in and promotes proliferation of melanoma cells. Our results provide mechanisms for HSF1 phosphorylation-dependent establishment of an active chromatin status, which is important for tumorigenesis.

The regulation of transcription by RNA polymerase II (Pol II) is associated with changes in the structure of chromatin[1]. The nucleosome core in chromatin is composed of 147 bp of DNA wrapped around a histone octamer and is one of the most stable protein–DNA complexes through multiple interactions between them. However, it is highly dynamic and tightly regulated by various protein complexes, including chromatin-remodeling complexes, and histone-modifying enzymes that catalyze post-translational modifications, such as acetylation, methylation, ADP-ribosylation, ubiquitination, and sumoylation. These histone modifications in the upstream regions and gene bodies change the net charge of nucleosomes or are recognized by reader proteins that affect chromatin dynamics[2], and regulate transcription initiation and elongation[1,3]. During the initial steps of gene activation,

transcription factors bind to specific DNA sequences near target genes and recruit transcriptional co-activators, including a number of histone-modifying enzymes, which establish an active chromatin state and facilitate the formation of a pre-initiation complex (PIC) containing Pol II on the promoter[4]. A single transcription factor may elicit different types of histone modifications via the recruitment of multiple protein complexes such as the SAGA histone acetyltransferase (HAT) complex and the Set1-containing methyltransferase complex COMPASS[5,6].

The heat shock response (HSR) is an evolutionarily conserved adaptive mechanism against proteotoxic stresses, including heat shock, which maintains cellular proteostasis[7,8], and is characterized by the induced expression of genes coding for a set of heat

[1]Department of Biochemistry and Molecular Biology, Yamaguchi University School of Medicine, Minami-Kogushi 1-1-1, Ube, Yamaguchi 755-8505, Japan. [2]Department of Omics and Systems Biology, Graduate School of Medical and Dental Sciences, Niigata University, Ichibancho 757, Asahimachi-dori, Chuo-ku, Niigata 951-8510, Japan. [3]Department of Molecular and Cellular Biology, Medical Institute of Bioregulation, Kyushu University, Maidashi 3-1-1, Higashi-ku, Fukuoka 812-8582, Japan. [4]Institute for Quantitative Biosciences, University of Tokyo, Tokyo 113-0032, Japan. ✉e-mail: anakai@yamaguchi-u.ac.jp

shock proteins (HSPs) and numerous non-HSP proteins[9]. This response is mainly regulated at the level of transcription by heat shock transcription factors (HSFs) that bind to the heat shock response elements (HSEs) in eukaryotes[10–12]. Among the four HSF family members in mammals, HSF1 is a master regulator of *HSP* genes. HSF1 exists in an equilibrium between an inert monomer and a DNA-binding trimer within cells, and mostly remains as a monomer via the negative regulation of chaperones, including HSP70 chaperones, under non-stressed condition[13]. On the other hand, a small amount of the HSF1 trimer constitutively binds to *HSP* promoters in complex with replication protein A and FACT[14], and also recruits the poly(ADP-ribose) polymerase PARP13–PARP1 complex[15]. These complexes allow for paused Pol II at the promoter-proximal region and a relatively open chromatin environment.

In response to heat shock, HSF1 is converted to a DNA-binding trimer by the release of chaperones, and its transcriptional activity is enhanced by post-translational modifications including phosphorylation at S230, S320, S326, and S419[16–19]. Activated and promoter-bound HSF1 facilitates the redistribution of PARP1 within a gene locus and recruits different types of co-activators, including components of Mediator and shugoshin 2 (SGO2), which directly facilitate the formation of PIC in a manner that is dependent on HSF1-S326 phosphorylation[20,21], transcription factors, such as ATF1 and PGC1α[22,23], a SWI/SNF chromatin-remodeling complex containing BRG1[24], and histone-modifying enzymes, including p300/CBP and GCN5 HATs[25–27], and the methyltransferase MLL1[28]. In *Drosophila*, HSF has been shown to recruit Mediator components[29], the histone methyltransferase Set1, and CBP, GCN5, and Tip60 HATs in *HSP70* promoter during heat shock[30–33]. Set1 is necessary for H3K4me3 in the promoter[32] and Tip60 is responsible for H2AK5ac and H4K5ac, which result in the activation of PARP and the exchange of H2A with its variant in the gene body[33,34]. In mammalian cells, active histone modifications in *HSP* promoters are also assumed to be enhanced in part by the enzymes described above. However, the mechanisms underlying histone modifications and the molecular relationships between different modifications in the mammalian HSR have not yet been elucidated in detail. These epigenetic mechanisms mediated by HSF1 are suggested to be closely associated with initiation and progression of cancer[35,36].

Here we employed chromatin immunoprecipitation combined with mass spectrometry (ChIP-MS) to comprehensively identify proteins cross-linked with chromatin in HSF1-binding regions[37]. This experiment highlights the enrichment of TRRAP, a scaffold protein of HAT complexes[38], and tripartite motif (TRIM) 33 and TRIM24, which are both readers of acetylated histone marks and the writers of the ubiquitin codes[39], predominantly under heat shock conditions. Our detailed analyzes reveal that HSF1 recruits the TRRAP-TIP60 HAT complex as well as p300, which establishes an active chromatin state through histone acetylation and acetylation-dependent H2B mono-ubiquitination by TRIM33 and TRIM24 in *HSP* promoters. The recruitment of these histone modifying enzymes is triggered by phosphorylation of HSF1-S419 via PLK1. Furthermore, this phosphorylation is constitutively enhanced in and promoted the proliferation of cancer cells.

## Results

### Identification of the co-activator TRRAP that interacts with HSF1

In search for factors that regulate epigenetic modifications in cooperation with HSF1 in human cells, we identified proteins that were coprecipitated with endogenous HSF1 at higher levels under heat shock conditions than under control conditions using ChIP-MS (Supplementary Fig. 1a and Supplementary Data 1). Among top 31 proteins out of the 236 identified proteins enriched upon heat shock, TRIM33, TRIM24, and TRRPA, as well as EP300 (p300) and CREBBP (CBP)

HATs[22,25], appeared to be associated with histone modifications (Fig. 1a, Supplementary Fig. 1B and Supplementary Data 2). We initially examined a role of TRRAP, a scaffold of the SAGA or TIP60 HAT complexes[38], and found that TRRAP was co-precipitated with HSF1 in nuclear extracts of heat-shocked cells (Fig. 1b). TRRAP knockdown (KD) reduced mRNA levels of HSP72 (HSPA1A) and other HSPs during heat shock (Fig. 1c and Supplementary Fig. 1c). TRRAP KD also reduced HSP72 mRNA expression during treatments with a proteasome inhibitor and a proline analog (Supplementary Fig. 1d). We further performed ChIP-seq analysis to identify the binding peaks of HSF1 and TRRAP. Although the numbers of identified TRRAP binding peaks were limited, they co-occupied 28 sites in heat-shocked cells, including many sites within the gene promoters of major HSPs, co-chaperone (p23), and ubiquitin (UBB) (Fig. 1d–f and Supplementary Fig. 1e, f). These results suggested that HSF1 promotes the HSR by recruiting the co-activator TRRAP in the promoters of *HSP* genes.

### Phosphorylated HSF1 at Ser419 recruits TRRAP to *HSP72* promoter

Since HSF1 is phosphorylated during heat shock[17], we examined its involvement in the interaction between HSF1 and TRRAP (Fig. 2a). We found that the majority of HSF1 phosphorylation site mutants, except for HSF1-S419 mutants, were co-precipitated with TRRAP in extracts from heat-shocked cells (Fig. 2a, b). We generated an antibody specific for phosphorylated HSF1 at S419 (Supplementary Fig. 2a), and showed that HSF1-S419 was phosphorylated at a low level under control conditions, while it was highly phosphorylated during treatments with heat shock or other proteotoxic stress inducers (Fig. 2c and Supplementary Fig. 2b). Constitutive and heat shock-induced S419-phosphorylated bands consistently disappeared after treatment with lambda protein phosphatase (Fig. 2d). The S419 is located in the DHR domain and is evolutionarily conserved in vertebrate HSF1 isoforms, but not in HSF2, HSF3, or HSF4 isoforms (Fig. 2e)[40]. Substitution of endogenous HSF1 with HSF1-S419A or HSF1-S419G reduced the expression of *HSP72* during heat shock (Fig. 2f), indicating that HSF1-S419 phosphorylation enhances its transcriptional activity. HSF1-S419 mutants translocated to the nucleus during heat shock (Supplementary Fig. 2c). TRRAP was recruited to *HSP72* promoter during heat shock, but was not recruited by HSF1 KD or substitution with HSF1-S419 mutants (Fig. 2g). The impaired recruitment of TRRAP was associated with a reduction in HSF1 binding and Pol II recruitment (Fig. 2g, h and Supplementary Fig. 2d). These results suggested that phosphorylated HSF1 at S419 recruits TRRAP to stabilize its transcription complexes and promote *HSP72* expression during heat shock.

We then identified a minimal region (amino acids 1801-1833) of TRRAP that was required for interaction with HSF1 (Supplementary Fig. 2e, f), and showed that HSF1 interacted with wild-type hTRRAP-3×FLAG, but not with the interaction mutant hTRRAPΔ1801-1833-3×FLAG (Supplementary Fig. 2g). Furthermore, substitution of endogenous TRRAP with the interaction mutant reduced *HSP72* expression during heat shock (Supplementary Fig. 2h). These results confirmed that HSF1-mediated TRRAP recruitment promotes *HSP72* expression.

### TRRAP–TIP60 complex is responsible for histone acetylation at specific residues

We hypothesized that a TRRAP-containing HAT complex acetylates histones in *HSP72* promoter. Indeed, pan-acetylated histone H3 (H3ac) and H4 (H4ac) levels were markedly elevated during heat shock, and these levels were only modestly increased by the substitution with hHSF1-S419 mutants (Fig. 3a and Supplementary Fig. 3a). Since TRRAP is a common subunit of several HAT complexes including TIP60 and GCN5[38], we identified proteins that were dominantly interacting with hTRRAP-3×FLAG during heat shock (Supplementary Fig. 3b and

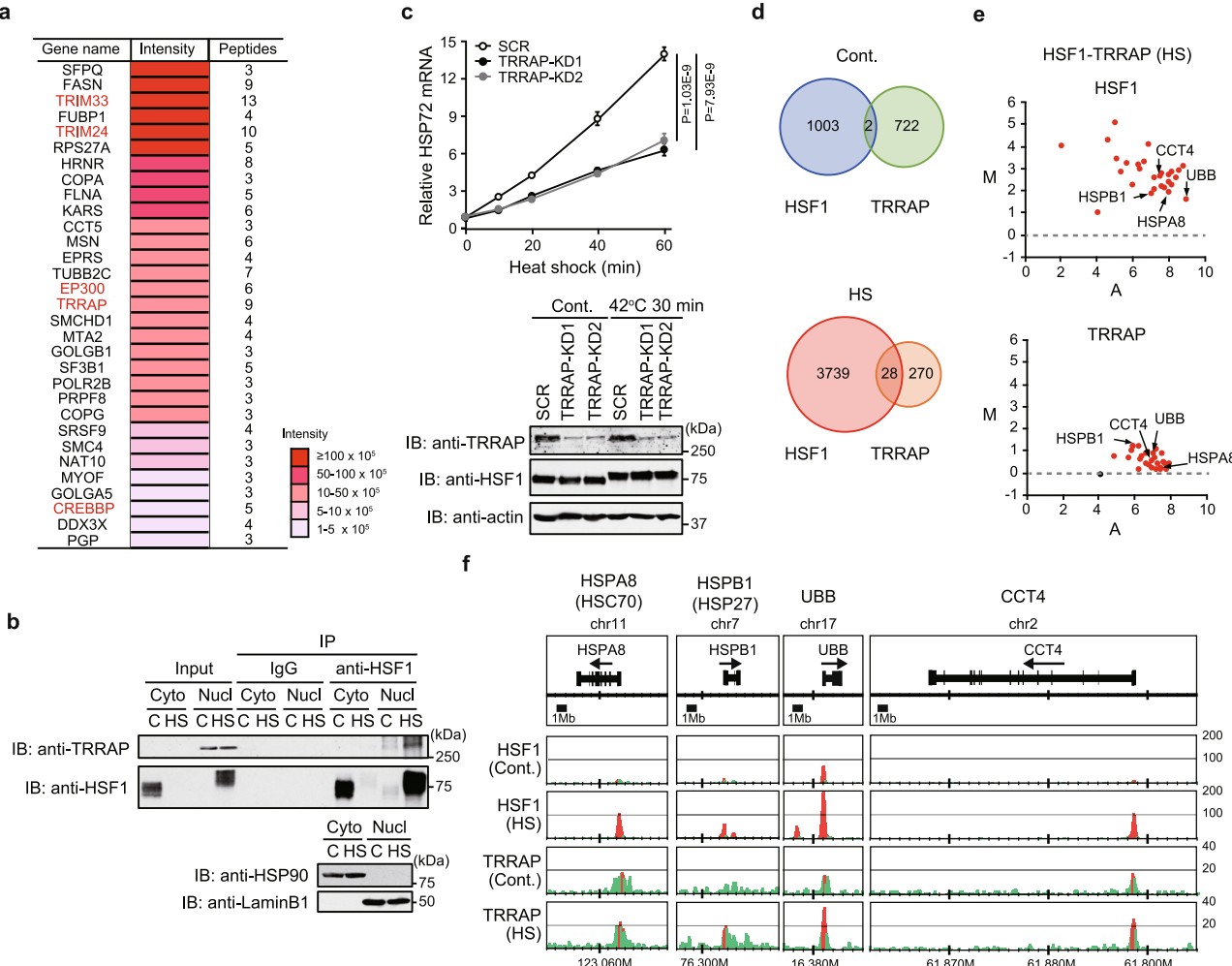

**Fig. 1 | Identification of the co-activator TRRAP that interacts with HSF1.**
**a** Relative enrichment of proteins identified in HSF1 ChIP preparations from heat-shocked cells. HeLa cells were untreated or treated with heat shock at 42 °C for 60 min, and HSF1-interacting proteins were identified by ChIP-MS. Thirty-one proteins highly enriched upon HS (difference of peptide numbers >3) are shown. Proteins related to histone modifications are indicated in red. **b** HSF1 interacts with TRRAP in the nucleus during heat shock. Cytoplasmic (Cyto) and nuclear (Nucl) extracts were prepared and complexes co-immunoprecipitated using anti-IgG or anti-HSF1 and subjected to immunoblotting. **c** Expression of HSP72 mRNA in TRRAP-KD cells during heat shock. Levels of HSP72 mRNA were quantified, and the levels relative to that in control SCR-treated cells are shown. Extracts of cells were subjected to immunoblotting. **d** Venn diagram of HSF1 and TRRAP ChIP-seq binding peaks in HeLa cells untreated (Cont.) or treated with

heat shock (HS). Because the TRRAP antibody generates a low signal-to-noise ratio in ChIP assay, the limited numbers of TRRAP binding peaks were identified using ChIP-seq. **e** MA (log ratio versus abundance) plot of ChIP-seq binding intensities for HSF1 and TRRAP in control ($R_1$) and heat shocked ($R_2$) cells at the common binding peaks for HSF1-HS/TRRAP-HS (28 peaks). The number of reads in the peak regions after normalization for a given sample was counted and the $M$ and $A$ values of each peak were calculated and plotted, where $M = \log_2(R_2/R_1)$ and $A = \log_2(R_1 + R_2)/2$. Dots with ratios ($M$) that increased during heat shock ($-\log_{10}P > 1$) are indicated in red. **f** ChIP-seq binding profiles of HSF1 and TRRAP in control (Cont.) and heat-shocked (HS) cells. Normalized read numbers are shown and peaks are indicated in red. Norminal $p$-values were determined by by two-way ANOVA in **c**. Error bars indicate SEM ($n = 3$) in **c**. Experiments were repeated two times for **b**.

Supplementary Data 3). These proteins included components of the TRRAP-p400-TIP60 (NuA4) complex (hereinafter referred to as the TRRAP-TIP60 complex), such as p400, ING3, GAS41, BAF53 and TIP60 HAT[41] (Fig. 3b), but did not include other HATs (Supplementary Data 3). We found that TIP60 and p400 were co-precipitated with HSF1 during heat shock (Fig. 3c). TRRAP KD reduced co-precipitation of TIP60 and p400, whereas TIP60 KD did not affect that of TRRAP and p400. ChIP assay showed that TIP60 occupied *HSP72* promoter during heat shock, but did not occupy it when HSF1 was KD or substituted with hHSF1-S419 mutants (Fig. 3d). Conversely, TIP60 KD moderately impaired the HSF1 binding to the promoter (Supplementary Fig. 3c). Furthermore, the KD of TIP60 or p400 reduced induction of HSP72 mRNA, but that of GCN5 or a related factor PCAF did not (Fig. 3e). These results indicated that HSF1 promotes *HSP72* expression by recruiting the TRRAP-TIP60 complex during heat shock.

Consistent with previous findings, p300 was also recruited to *HSP72* promoter during heat shock (Fig. 1a), and was still recruited at reduced levels upon TRRAP KD (Supplementary Fig. 3d). The KD of both p300 and TRRAP (DKD) induced greater reductions in HSP72 mRNA levels than TRRAP KD alone, suggesting non-redundant functions (Supplementary Fig. 3e). We therefore investigated prevalent histone acetylation marks (H3K9ac, H3K27ac, and H4K16ac) and the marks recognized by TRIM33 (H3K18ac) and TRIM24 (H3K23ac) in *HSP72* promoter[15,39]. The KD of TRRAP or TIP60 abolished H3K18ac and H4K16ac marks upon heat shock (Fig. 3f and Supplementary Fig. 3f). TIP60-mediated H3K18ac was unexpected because it is mediated by p300 in the promoters of activated genes during antiviral and hormonal responses[42,43]. In contrast, p300 KD abolished H3K23ac and H3K27ac marks[43] (Fig. 3f). It is important to note that H3K9ac mark was not blocked by the KD of TIP60 or p300 (Supplementary Fig. 3g). Thus,

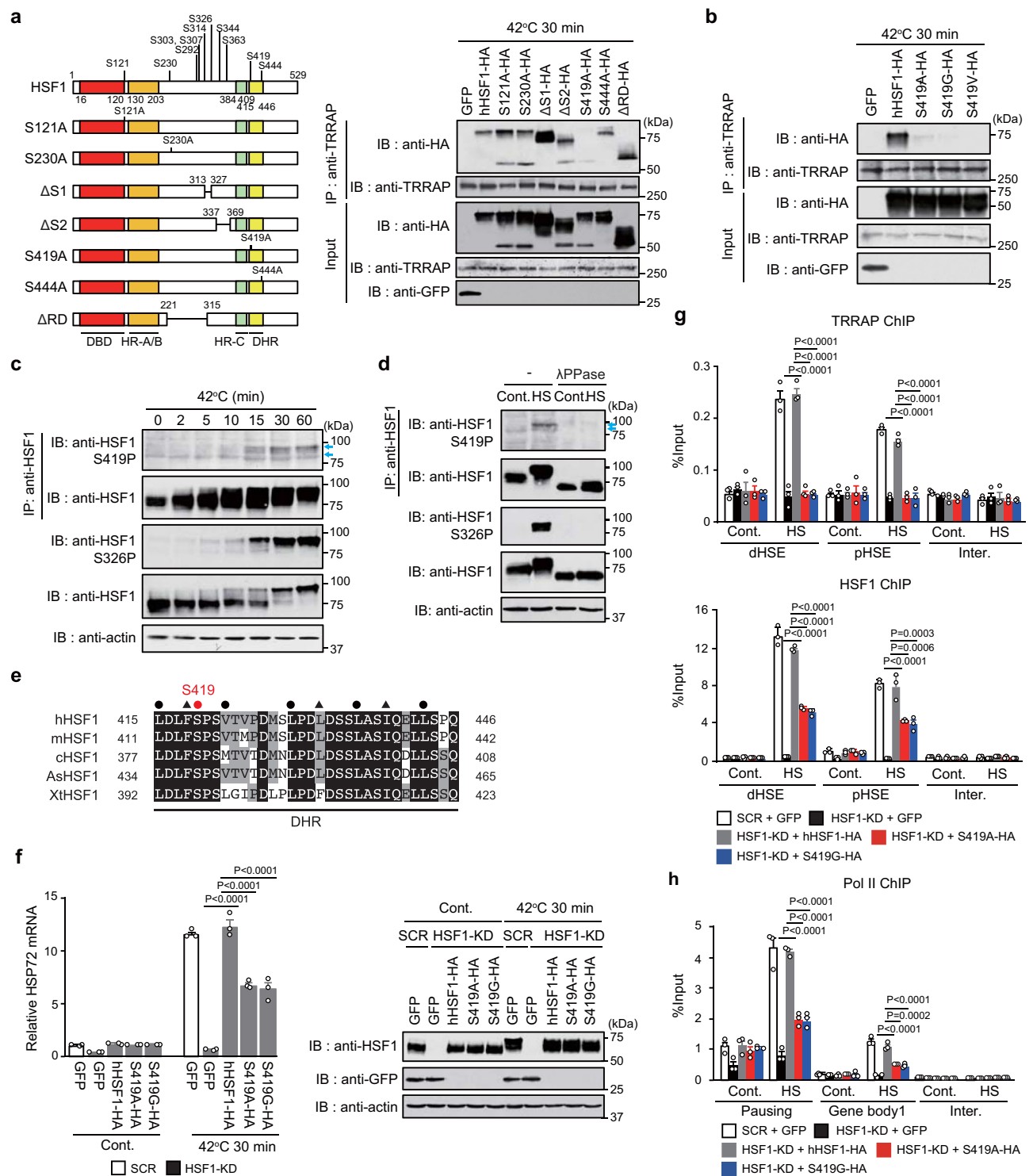

**Fig. 2 | Phosphorylated HSF1 at S419 recruits TRRAP to *HSP72* promoter.**
**a, b** Schematic representation of phosphorylation sites in hHSF1 and its
mutants. DBD (red box), DNA-binding domain; HR (orange box), hydrophobic
heptad repeat; DHR (light green box), downstream of HR-C (yellow box).
Extracts of heat-shocked HSF1-null MEF cells expressing these mutants or GFP
were co-immunoprecipitated using anti-TRRAP and subjected to immunoblot-
ting. HeLa cells were treated with heat shock, and cell extracts were subjected to
HSF1 immunoprecipitation and immunoblotting using antibody for HSF1
phospho-S419 or HSF1 (**c**). Some extracts were incubated without (−) or with
lambda protein phosphatase (λPPase) (**d**). Blue arrows indicate the positions of
HSF1-S419 phosphorylated bands. The intensity of the upper band was markedly
enhanced during heat shock. **e** Alignment of amino acid sequences for the DHR

domain containing S419 in human, mouse, chicken, lizard (*Anolis sagrei*, As),
and frog (*Xenopus tropicalis*, Xt) HSF1. This domain is characterized by the
heptad repeats of hydrophobic amino acids (black circles and triangles), which
are conserved in vertebrate HSF family members (HSF1 to HSF4). A red circle
indicates the position of HSF1-S419. **f** Cells, in which endogenous HSF1 was
replaced with hHSF1-HA, S419A-HA or S419G-HA, were heat-shocked, and HSP72
mRNA levels were quantified. Extracts from these cells were subjected to
immunoblotting. Occupancy of TRRAP, HSF1 (**g**), or Pol II (**h**) in cells expressing
HSF1-S419 mutants. Cells were treated as described in **f**, and ChIP assay was
performed. d, distal; p, proximal; and inter, intergenic. Norminal *p*-values were
determined by one-way ANOVA, followed by Tukey-Kramer test in **f**–**h**. Error
bars indicate SEM (*n* = 3) in **f**–**h**. Experiments were repeated two times for **a**–**d**.

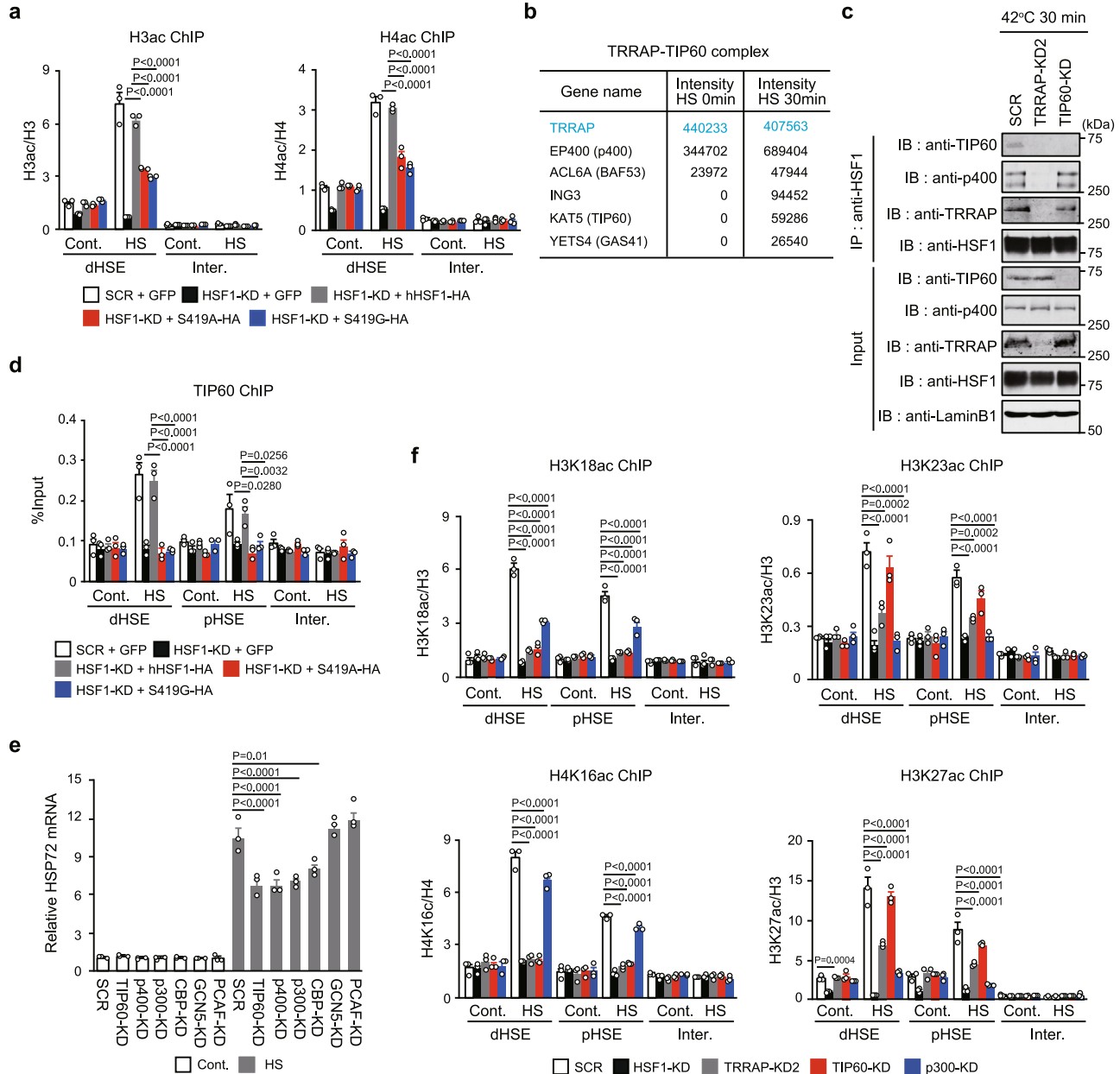

**Fig. 3 | TRRAP-TIP60 complex is responsible for histone acetylation at specific residues. a** Occupancy of pan-acetylated histone H3 and H4 in *HSP72* promoter. ChIP assay was performed using HeLa cells treated as described in Fig. 2f. **b** Components of the TRRAP-TIP60 complex identified dominantly in heat-shocked cells. Nuclear extracts were prepared from cells overexpressing hTRRAP-3 × FLAG, and proteins co-immunoprecipitated with anti-FLAG were identified by MS. **c** Interaction between HSF1 and TIP60. Complexes co-immunoprecipitated using anti-HSF1 in nuclear extracts of heat-shocked cells were subjected to immunoblotting. **d** Occupancy of TIP60 in cells expressing HSF1-S419 mutants. ChIP assay was performed using cells treated as described in **a**. **e** Cells, in which components of HAT complexes were knocked down, were treated with heat shock. Levels of HSP72 mRNA were quantified, and relative levels are shown. **f** Occupancy of active chromatin marks in *HSP72* promoter. ChIP assay was performed using untreated (Cont.) or heat-shocked (HS) cells, in which HSF1, TRRAP, TIP60, or p300 were knocked down. Norminal *p*-values were determined by one-way ANOVA, followed by Tukey-Kramer test in **a** and **d**–**f**. Error bars indicate SEM (*n* = 3) in **a** and **d**–**f**. Experiments were repeated two times for **c**.

the TRRAP-TIP60 complex is responsible for H3K18ac and H4K16ac marks during heat shock, whereas p300 is required for H3K23ac and H3K27ac.

## Acetylation-dependent histone H2B mono-ubiquitination by TRIM33

TIP60- and p300-mediated acetylation marks may be recognized by the reader proteins TRIM33 and TRIM24 (Fig. 1a), so we investigated genome-wide occupancy of them. ChIP-seq analysis showed that HSF1, TRIM33, and TRIM24 co-occupied 1145 sites in heat-shocked cells (Fig. 4a, b). Among the 28 HSF1-TRRAP co-occupied sites upon heat

shock, 24, which were located within the gene promoters of major HSPs and UBB, were also occupied by both TRIM33 and TRIM24 (Fig. 4b and Supplementary Fig. 4a). Even in control cells, 925 out of the 1005 HSF1-binding sites (92%) were co-occupied by them, suggesting a strong relationship between HSF1 and the two TRIM proteins. Indeed, the KD of TRIM33 or TRIM24 reduced the expression levels of HSP72 mRNA during heat shock (Fig. 4c).

TRIM33 and TRIM24 dominantly localized in chromatin fraction (Supplementary Fig. 4b), and were co-precipitated with HSF1 in nuclear extracts of heat-shocked cells (Fig. 4d). We examined effects of reduced histone acetylation on the interaction between the TRIM proteins and

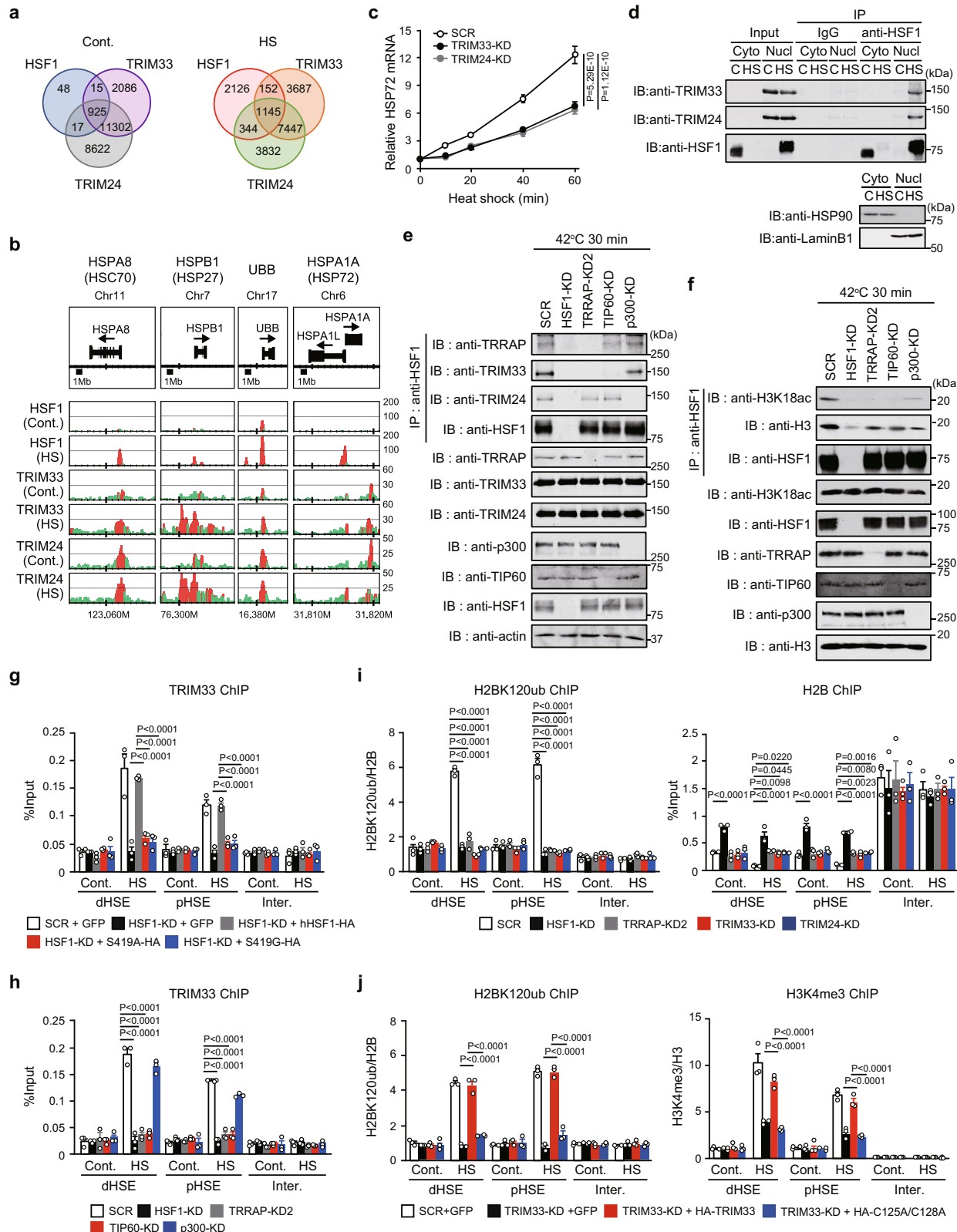

HSF1, and found that TRRAP KD or TIP60 KD reduced co-precipitation of the H3K18ac reader protein TRIM33 with HSF1, but did not affect that of TRIM24 (Fig. 4e). As expected, the same treatment reduced co-precipitation of histone H3K18ac with HSF1 in extracts from cells cross-linked with DSP (Fig. 4f). These results suggested that histone H3K18ac stabilizes the interaction between TRIM33 and HSF1. Similarly, p300 KD reduced co-precipitation of the H3K23ac reader protein TRIM24 with

HSF1 (Fig. 4e). We then examined effects of HSF1 phosphorylation and histone acetylation on the recruitment of TRIM33 and TRIM24 using ChIP assay. TRIM33 was recruited to *HSP72* promoter during heat shock; however, its recruitment was blocked by HSF1 KD or its substitution with HSF1-S419 mutants (Fig. 4g). Furthermore, the KD of TRRAP or TIP60, but not p300 KD, blocked the recruitment of TRIM33 (Fig. 4h). In contrast, the recruitment of TRIM24 was only modestly reduced by the

**Fig. 4 | Acetylation-dependent histone H2B mono-ubiquitination by TRIM33.**
Venn diagram of HSF1, TRIM33, and TRIM24 ChIP-seq binding peaks (**a**) and their
binding profiles (**b**) in control (Cont.) and heat-shocked (HS) HeLa cells. Normalized
read numbers are shown (green), and peaks are indicated in red. **c** Expression of
HSP72 mRNA in TRIM33- or TRIM24-KD cells during heat shock. HSP72 mRNA levels
relative to that in control SCR-treated cells are shown. **d** HSF1 interacts with TRIM33
and TRIM24 in the nucleus during heat shock. Cytoplasmic (Cyto) and nuclear
(Nucl) extracts were prepared and complexes co-immunoprecipitated using anti-
IgG or anti-HSF1 were subjected to immunoblotting. **e** Nuclear extracts were pre-
pared from heat-shocked cells, in which HSF1, TRRAP, TIP60, or p300 was knocked
down, and complexes co-immunoprecipitated using anti-HSF1 were subjected to
immunoblotting. **f** HSF1 interacts with histone H3K18ac. Cells treated as described
in **e** were incubated with the crosslinking reagent DSP. Nuclear extracts were

prepared and co-immunoprecipitated complexes were subjected to immunoblot-
ting. **g** TRIM33 occupancy in cells expressing HSF1-S419 phosphorylation site
mutants. Cells were treated as described in Fig. 2f, and ChIP assay was performed. **h** TRIM33 occupancy in TRRAP- or TIP60-KD cells. Cells were treated as described
in **e**. **i** TRIM33- and TRIM24-dependent H2BK120ub in *HSP72* promoter. Cells,
infected with adenovirus expressing shRNA for HSF1, TRRAP, TRIM33, or TRIM24,
were heat-shocked, and ChIP assay was performed. **j** Occupancy of H2BK120ub
and H3K4me3 in cells expressing a TRIM33 mutant. Cells, in which endogenous
TRIM33 was replaced with the RING domain mutant of HA-hTRIM33, were heat-
shocked, and ChIP assay was performed. Norminal *p*-values were determined by
two-way ANOVA in **c** or by one-way ANOVA, followed by Tukey-Kramer test in
**g–j**, Error bars indicate SEM (n = 3) in **g–j**. Experiments were repeated two times for
**d**, **e**, and **f**.

substitution with HSF1-S419 mutants or the KD of TRRAP or TIP60;
however, it was blocked by p300 KD (Supplementary Fig. 4c, d). These
results indicated that HSF1-S419 phosphorylation and histone acetyla-
tion are required for the recruitment of both TRIM33 and TRIM24 in
*HSP72* promoter during heat shock.

Since TRIM37 has been shown to mono-ubiquitinate histone
H2A[44], we examined the involvement of TRIM33 and TRIM24 in mono-
ubiquitination of histone H2B on Lys120 (H2BK120ub), which is asso-
ciated with an active chromatin state[45]. The KD of TRIM33 or
TRIM24 reduced H2BK120ub levels during heat shock (Supplementary
Fig. 4e). ChIP assay showed that H2BK120ub levels were markedly
elevated at the HSEs in the *HSP72, HSP27, HSP40, and HSP110* pro-
moters during heat shock, but were not increased by HSF1 KD (Fig. 4i
and Supplementary Fig. 4f). Furthermore, these levels in *HSP72* pro-
moter were not elevated during heat shock by the KD of TRRAP,
TRIM33, or TRIM24 (Fig. 4i). Increases in H2BK120ub levels were also
blocked by the substitution of endogenous TRIM33 with the RING
domain mutant HA-hTRIM33-C125A/C128A (Fig. 4j and Supplementary
Fig. 4g). Consistent with the cross-talk between H2BK120ub and the
active chromatin mark H3K4me3[46], increases in H2BK120ub levels
were associated with elevated H3K4me3 levels, which were suppressed
by the substitution with the RING domain mutant (Fig. 4j and Sup-
plementary Fig. 4g). H2BK120ub within gene bodies, which is coupled
with transcription elongation, was catalyzed by the RNF20/RNF40 E3
ubiquitin ligase complex[47]. The increases observed in H2BK120ub
levels in *HSP72* gene body regions upon heat shock were blocked by
the KD of RNF20 or RNF40, but was not by that of TRIM33 (Supple-
mentary Fig. 4h). These results demonstrated that TRIM33 and TRIM24
are required for acetylation-dependent histone H2BK120ub in *HSP72*
promoter during heat shock.

### PLK1 phosphorylates HSF1-S419 and triggers the establishment of an active chromatin state

We then searched for protein kinases that phosphorylate HSF1-S419
upon heat shock. Three protein kinases, PLK1, CSNK1A1, and NEK7,
were identified as HSF1-interacting proteins by ChIP-MS (Supplemen-
tary Fig. 1b and Supplementary Data 1). Therefore, we examined their
involvement and found that the KD of PLK1, but not others, blocked
S419 phosphorylation under both control and heat shock conditions
(Fig. 5a). This result was consistent with previous findings showing that
HSF1 interacts with PLK1 in vivo and this interaction increases during
heat shock, and PLK1 can phosphorylates S419 in vitro and positively
influence HSF1 nuclear accumulation[18]. The S419 phosphorylation was
strongly inhibited in HeLa and MeWo cells treated with the PLK1
inhibitors BI6727 and BI2536, but was not inhibited by treatment with
an inhibitor of ERK, MEK, or mTOR (Fig. 5b and Supplementary Fig. 5a).
In contrast, phosphorylation of S326, one of important phosphory-
lated serine residues, was partially inhibited by treatments with these
inhibitors[48–50], except for the PLK1 inhibitor. Phosphorylation of S419
as well as S326 was found to have a great impact on the expression of
HSP72 mRNA during heat shock (Supplementary Fig. 5b). Thus, PLK1

phosphorylates HSF1-S419 under normal conditions, and its phos-
phorylation is induced during heat shock.

PLK1 was not only maximally expressed in the G2/M phase of the
cell cycle but also expressed at low levels in the G1 and S phases
(Supplementary Fig. 5c)[51], and was dominantly localized in chromatin
fraction (Supplementary Fig. 5d). It was highly co-precipitated with
HSF1 in nuclear extracts from heat-shocked cells (Fig. 5c), and was
recruited to *HSP72* promoter during heat shock in a manner dependent
on HSF1 (Supplementary Fig. 5e). We then substituted endogenous
PLK1 with the kinase-dead mutant PLK1-K82R. This mutant interacted
with HSF1 (Fig. 5d), but did not phosphorylate S419 during heat shock
(Fig. 5e). HSF1 continued to bind to *HSP72* promoter in heat-shocked
cells expressing PLK1-K82R, although its binding levels were reduced
(Fig. 5f). Importantly, TRRAP and TRIM33 were not recruited to *HSP72*
promoter in the same cells. Furthermore, the impaired recruitment of
TRIM33 was associated with blockade of H2BK120ub (Fig. 5g). These
results indicated that PLK1-mediated HSF1-S419 phosphorylation trig-
gers the establishment of an active chromatin state and stabilizes its
transcription complexes in *HSP72* promoter during heat shock.

### HSF1-S419 phosphorylation maintains proteostasis capacity

We investigated whether phosphorylation of HSF1-S419 is associated
with proteostasis capacity. We found that cell survival was markedly
reduced by TRRAP KD or the substitution with hHSF1-S419 mutants
during extreme heat shock (Fig. 6a, b). The reduction in cell survival
was accompanied by the greater accumulation of insoluble ubiquity-
lated proteins within cells (Fig. 6c). We performed luciferase refolding
assays, and found that luciferase refolding was severely impaired by
TRRAP KD or the substitution with hHSF1-S419 mutants (Fig. 6d, e).
These results demonstrated that phosphorylation of HSF1-S419 main-
tains proteostasis capacity during heat shock.

### HSF1 phosphorylation supports tumor formation

Since HSF1 is activated to cope with chronic proteotoxic stress in
cancer cells[52] and proliferation of melanoma cells is strongly depen-
dent on HSF1[14], we investigate the impact of S419 phosphorylation on
cancer cell proliferation. We found that levels of constitutive S419
phosphorylation were higher in various human melanoma cell lines
(MeWo, HMV-1, MMAc) than that in immortalized cells (OUMS-36T)
(Fig. 7a). Overexpression of PLK1 and its active mutant PLK1-T210D
increased S419 phosphorylation levels in immortalized cells (Fig. 7b).
Remarkably, the substitution with hHSF1-S419 mutants reduced the
proliferation of these melanoma cell lines by ~50%, but did not affect
that of immortalized cells (Fig. 7c). The substitution with S419 mutants
also partially reduced proliferation of various types of cancer cells
(Supplementary Fig. 6a). It is important to note that proliferation
reduction rates by HSF1 KD in melanoma cell lines were the highest
among those in other types of cancer cell lines (Supplementary Fig. 6b).

We next treated MeWo cells with the MEK inhibitor trametinib,
which suppresses cell proliferation and inhibits S326 phosphoryla-
tion, and found that the reduction observed in proliferation of

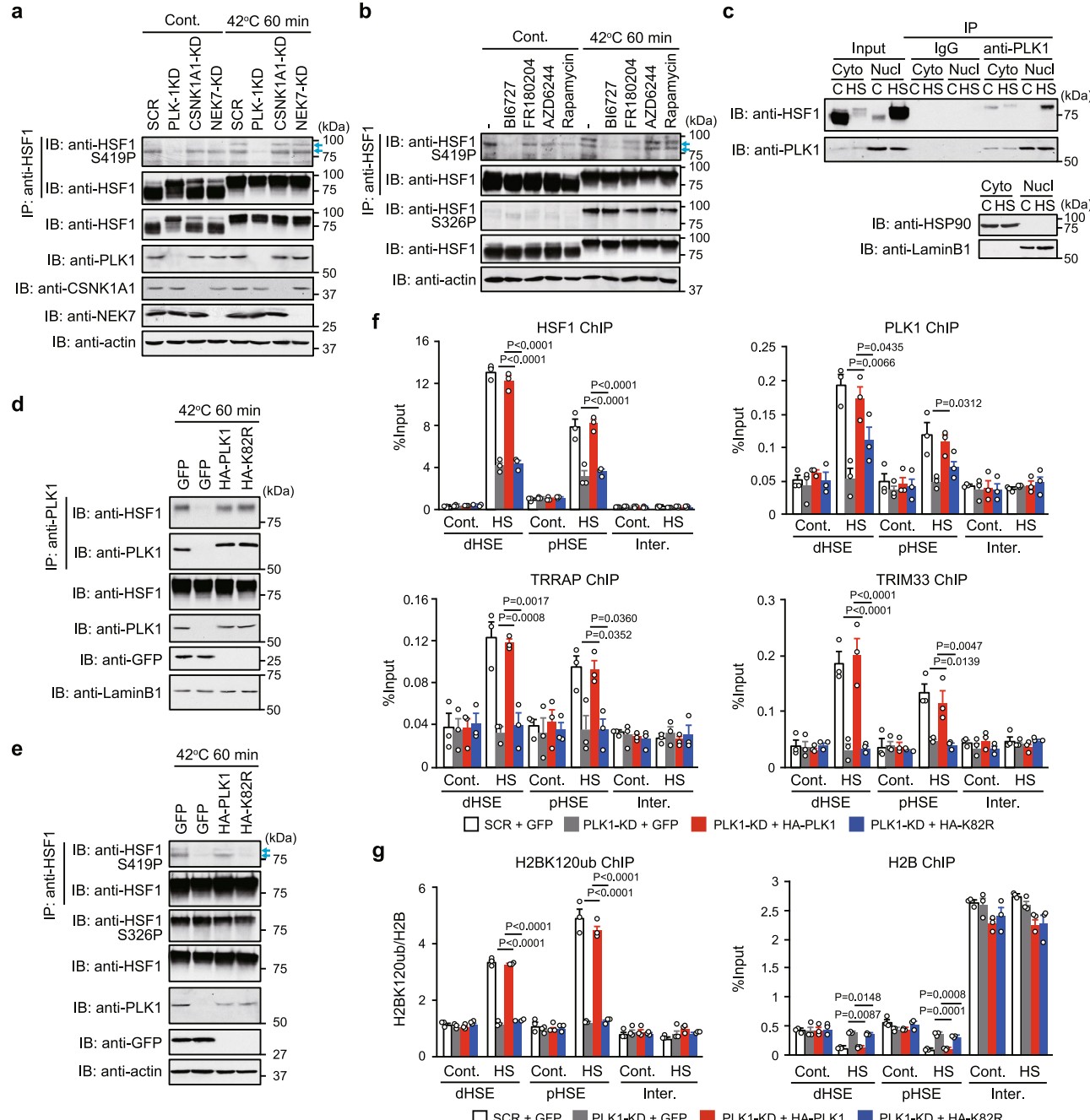

**Fig. 5 | PLK1 phosphorylates HSF1-S419 and triggers histone modifications.**
HeLa cells were infected with adenovirus expressing shRNA for PLK1, CSNK1A1, NEK7 or SCR (**a**), or treated with BI6727 (PLK1 inhibitor), FR180204 (ERK1/2 inhibitor), AZD6244 (MEK1/2 inhibitor), or rapamycin (mTOR inhibitor) (**b**). These cells were then treated with heat shock. Cell extracts were subjected to HSF1 immunoprecipitation and immunoblotting using antibody for HSF1 phospho-S419 or HSF1. Blue arrows indicate the positions of HSF1-S419 phosphorylated bands. The intensity of the upper band was markedly enhanced during heat shock. **c** PLK1 interacts with HSF1 in the nucleus during heat shock. Cell extracts were prepared as described in Fig. 1b and subjected to immunoblotting. Cells, in which endogenous PLK1 was replaced with GFP, wild-type HA-hPLK1, or its kinase-dead mutant, were treated with heat shock. Nuclear extracts were prepared and complexes co-immunoprecipitated using anti-PLK1 (**d**) or anti-HSF1 (**e**) were subjected to immunoblotting. Cells, in which endogenous PLK1 was replaced with GFP, HA-hPLK1, or HA-hPLK1-K82R, were treated with heat shock. ChIP-qPCR of HSF1, TRRAP, TRIM33, and PLK1 (**f**), or H2BK120ub and histone H2B (**g**) was performed. Norminal *p*-values were determined by one-way ANOVA, followed by Tukey-Kramer test in **f** and **g**. Error bars indicate SEM (*n* = 3) in **f** and **g**. Experiments were repeated two times for **a**–**e**.

hHSF1-S419-expressing cells was reinforced by treatment with trametinib (Supplementary Fig. 6c, d). Furthermore, substitution with the double mutant hHSF1-S326A/S419A reduced cell proliferation much more (75% reduction) than that with hHSF1-S326A or hHSF1-S419A (Fig. 7d). Re-expression of hHSF1-S326A/S419A did not restore the constitutive expression of HSP72 mRNA in HSF1-

depleted cells, whereas that of hHSF1-S326A or hHSF1-S419A partially restored it (Fig. 7e). We then performed xenograft experiments on athymic nude mice subcutaneously injected with MeWo cells, in which endogenous HSF1 was substituted with each mutant. Tumor formation by MeWo cells expressing hHSF1-S326A or hHSF1-S419A was significantly less than that by cells expressing wild-type

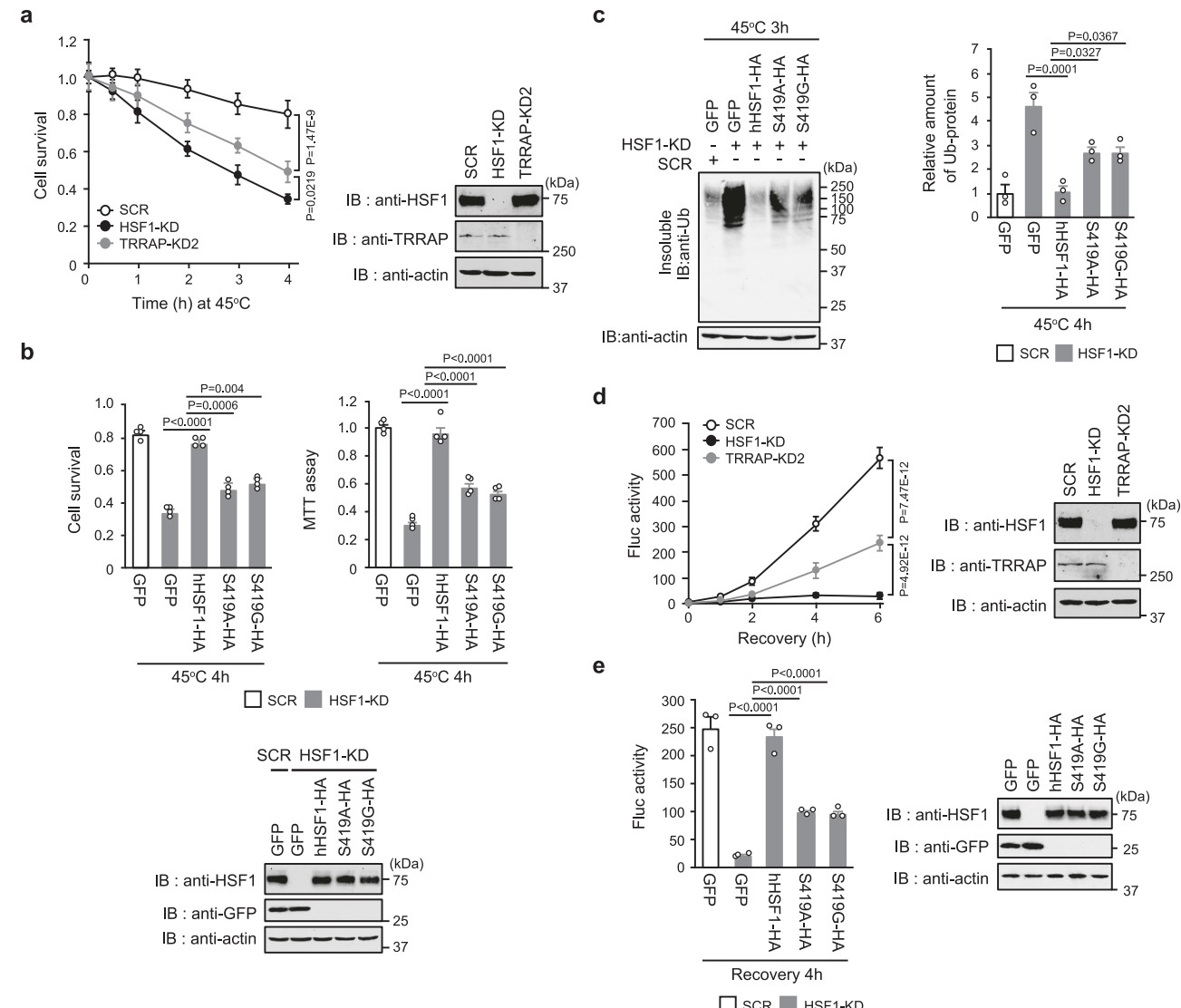

**Fig. 6 | HSF1-S419 phosphorylation maintains proteostasis capacity. a** TRRAP promotes cell survival. Cells were infected with adenovirus expressing shRNA for HSF1, TRRAP or SCR, and were heat-shocked at 45 °C for the indicated periods. Number of viable cells excluding trypan blue was counted. Extracts of cells were subjected to immunoblotting. **b** Phosphorylation of HSF1-S419 promotes cell survival. Cells, in which endogenous HSF1 was replaced with GFP, wild-type hHSF1-HA, or hHSF1-S419A mutants were heat-shocked at 45 °C for 4 h. Number of viable cells excluding trypan blue was counted, and MTT assay was performed. Extracts of cells were subjected to immunoblotting. **c** Phosphorylation of HSF1-S419 inhibits the accumulation of ubiquitylated proteins. Cells treated as described in **b** were heat-shocked at 45 °C for 4 h. The accumulation of insoluble ubiquitylated proteins was examined by immunoblotting using anti-Ub antibody and quantified. β-Actin levels in the soluble fraction were also shown. **d** TRRAP

promotes refolding of the luciferase sensor protein during recovery from heat shock. HeLa-Fluc cells treated as described in **a** were heat-shocked at 42 °C for 2 h. These cells were then recovered at 37 °C for the indicated periods, and luciferase activity values were calculated and normalized to the value of control cells (100%). Extracts of cells were subjected to immunoblotting. **e** Phosphorylation of HSF1-S419 promotes refolding of the luciferase sensor protein. HeLa-Fluc cells treated as described in **b** were heat-shocked at 42 °C for 2 h. These cells were then recovered at 37 °C for 4 h, and luciferase activity values relative to that of control cells were estimated. Extracts of cells were subjected to immunoblotting. Norminal $p$ values were determined by two-way ANOVA in **a** and **d** or by one-way ANOVA, followed by Tukey-Kramer test in **b**, **c** and **e**. Error bars indicate SEM ($n = 4$) in **a** and **b**, or ($n = 3$) in **c**, **d**, and **e**.

HSF1; importantly, tumor formation by hHSF1-S326A/S419A-expressing cells was markedly less than that by single mutant-expressing cells (Fig. 7f, g and Supplementary Fig. 6e). These results demonstrated that HSF1-S419 phosphorylation supports tumor formation by melanoma cells, and suggested the strong impact of HSF1-S326/S419 phosphorylation on melanoma cell proliferation.

## Discussion

Histone acetylation at multiple lysine residues is closely associated with active chromatin states, and is induced by a number of HAT complexes in response to stimuli. Upon heat shock, p300 and CBP

HATs are recruited to the promoters of *HSP* genes in mammalian cells[22,25,26]. However, limited information is available on the recruitment of other HAT complexes in *HSP72* promoter and their roles in the mammalian HSR[53]. Here, we demonstrated that HSF1 recruits the TRRAP-TIP60 HAT complex as one of coactivator complexes in a manner that is dependent on the phosphorylation of HSF1-S419 during heat shock (Figs. 1–3, 7h). TIP60 belongs to the MYST family of HATs comprising TIP60, MOZ, MORF, HBO1, and MOF[54], and can acetylate histone H2A, H3, and H4 in vitro[55]. It facilitates inducible gene expression in response to environmental and developmental cues, such as serum stimulation and hormone exposure, through the

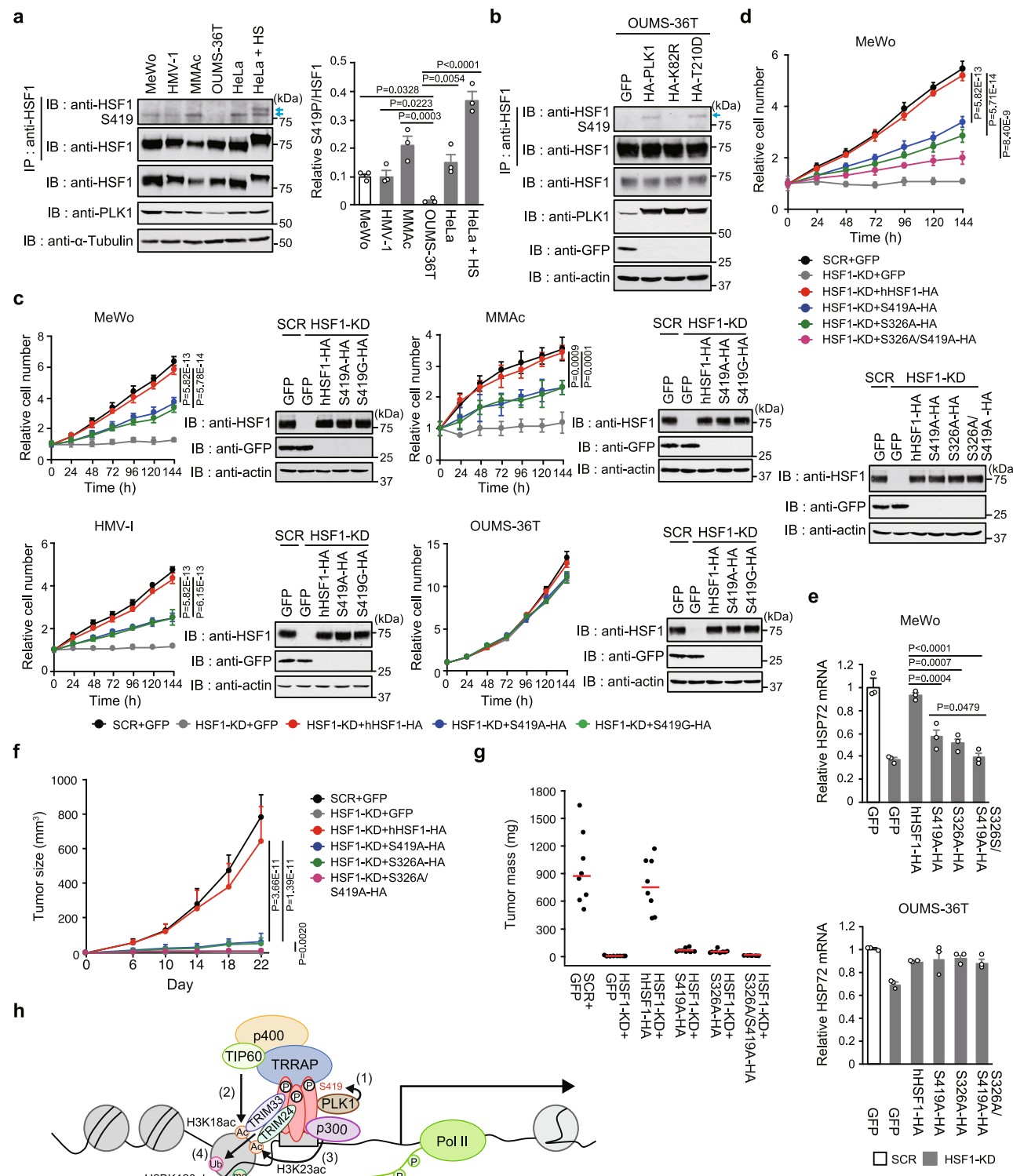

**Fig. 7 | HSF1 phosphorylation supports tumor formation. a** Equal amounts of total cell extracts from melanoma cell lines, HeLa cells (control and heat-shocked), and OUMS-36T-3F cells were subjected to HSF1 immunoprecipitation and immunoblotting. Relative levels of HSF1-S419 phosphorylation normalized by HSF1 protein levels are shown. **b** Phosphorylation of HSF1-S419 is induced by overexpression of PLK1 in OUMS-36T-3F cells. HSF1 immunoprecipitation and immunoblotting were performed as described in Fig. 5a. **c, d** Endogenous HSF1 was replaced with GFP, hHSF1-HA, S419A-HA, S419G-HA (**c, d**), or S326A/S419A-HA (**d**) in melanoma and OUMS-36T-3F cells. Cell extracts were subjected to immunoblotting. **e** MeWo and OUMS-36T-3F cells were treated as described in **d**. HSP72 mRNA levels were quantified and shown. **f, g** Tumor sizes and masses of melanoma cells

expressing hHSF1 phosphorylation site mutants in athymic nude mice. The sizes (**f**) and masses (**g**) of tumors at indicated time points after injection were calculated until 22 days. Bars in **g** indicate mean values ($n = 8$). **h** Schematic model for establishing an active chromatin state in *HSP72* promoter. PLK1 phosphorylates HSF1-S419 (1), which recruits the TRRAP-TIP60 complex. TIP60 and p300 are responsible for H3K18ac and H3K23ac, respectively (2, 3). TRIM33 and TRIM24, recruited to the promoter by interactions with HSF1 and the histone acetylation marks, are required for H2BK120ub (4). Norminal *p* values were determined by one-way ANOVA, followed by Tukey-Kramer test in **a** and **e** or by two-way ANOVA in **c, d** and **f**. Error bars indicate SEM ($n = 3$) in **a** and **e**, ($n = 4$) in **c** and **d**, or ($n = 8$) in **f**. Experiments were repeated two times for **b**.

acetylation of histone H2A and H4 in gene promoters[56,57]. The present results revealed that the TRRAP–TIP60 complex is responsible for the maximal acetylation of histone H3 and H4 (Fig. 3a), the stabilization of the HSF1-transcription complexes on *HSP72* promoter (Fig. 2g, h), and the inducible expression of *HSP* genes in the HSR (Figs. 1c, 2f, and 3e).

Acetylation of histones may directly contribute to the establishment of active or decondensed chromatin states through biophysical changes, including a change in the net charge of nucleosomes[2]. Previous studies reported that the intra-molecular folding of nucleosome arrays and the formation of compact 30-nm fibers in vitro are inhibited by the random hyperacetylation of histones[58] or even by the prevalent acetylation mark H4K16ac[59]. The acetylation of H4K16 is mediated by the MYST family members through the highly conserved MYST HAT domain[54]. Actually, TIP60 was responsible for H4K16ac in *HSP72* promoter during heat shock (Fig. 3f), suggesting that it directly facilitates the establishment of a decondensed chromatin state. More importantly, histone acetylation marks are recognized by reader proteins that affect chromatin structure. We showed that TIP60 is required for another mark H3K18ac (Fig. 3f). It was reported previously that a nodal stimulation of embryonic stem cells promotes formation of the Smad2/3 transcription factor-TRIM33 complex and the accessibility of the nodal response element to Smad2/3-Smad4 through the binding of TRIM33 to H3K18ac[60]. Similarly, TRIM33 was recruited to *HSP72* promoter during heat shock through its interaction with both HSF1 and TIP60-mediated H3K18ac (Fig. 4a–h). Remarkably, promoter-bound TRIM33 acted as an RING E3 ubiquitin ligase and was responsible for H2BK120ub at *HSP72* promoter (Fig. 4i), indicating the crosstalk between histone acetylation and ubiquitination in transcriptional activation[61]. Mono-ubiquitination at this lysine residue may directly interfere with the compaction of 30-nm chromatin fibers and inter-fiber interactions, and thereby leading to an accessible chromatin structure in vitro[62]. Furthermore, COMPASS and Dot1L histone methyltransferase complexes may be stabilized on chromatin by H2BK120ub and lead deposition of the other active histone marks H3K4me3 and H3K79me3, respectively[46,63]. Indeed, MLL1, a methyltransferase subunit of the COMPASS complex, is recruited to *HSP72* promoter in response to proteotoxic stress[28], and TRIM33-mediated H2BK120ub facilitated the deposition of H3K4me3 during heat shock (Fig. 4j). The present results revealed that the acetylation-dependent ubiquitination of histones in *HSP72* promoter plays an important role in the establishment of active chromatin states during the HSR, and suggest crosstalk between histone modifications, including acetylation, ubiquitination, and methylation, in these processes.

TRIM E3 ubiquitin ligase family proteins are characterized by a conserved N-terminal TRIM motif containing a RING domain and a variable C-terminal region, and grouped into 11 classes based on their C-terminal domain identities[64]. Class VI TRIM proteins comprising TRIM24, TRIM28, and TRIM33 uniquely function as chromatin-associated transcriptional co-regulators via the C-terminal domain containing a PHD-bromodomain module[39]. TRIM24, similar to TRIM33, acts as a co-activator of estrogen receptor α by stabilizing its interaction with chromatin through the recognition of H3K23ac[65]. Importantly, TRIM24 was also recruited to *HSP72* promoter during heat shock in a manner that was dependent on HSF1 and p300 (Fig. 4a-f and Supplementary Fig. 4c, d). Although TRIM24 and TRIM33 may form a hetero-dimer[66], each one was recruited to *HSP72* promoter independently of the other (Fig. 4h and Supplementary Fig. 4d). Nevertheless, both TRIM24 and TRIM33 were both needed for the mono-ubiquitination of H2BK120 (Fig. 4i). TRIM proteins generally require the dimerization of RING domains for E3 ligase activity[67], but none of the class VI RING domains forms a homodimer[68]. Our observations suggested that the stabilization of TRIM33 and TRIM24 on chromatin through interaction with both HSF1 and histone acetylation marks is necessary for RING dimerization and activity. Interestingly, another class VI protein TRIM28 was highly enriched in the HSF1 ChIP

preparation under both non-stressed and heat shock conditions (see ChIP-MS data). TRIM28 has been shown to constitutively stabilize paused Pol II, and facilitate its release for elongation upon heat shock[69]. Therefore, the class VI TRIM proteins are important for the regulation of the HSR.

Transcriptional activity of HSF1 is enhanced by phosphorylation of several serine residues in the HSR. For example, HSF1-S326 was phosphorylated by various protein kinases, including MEK1/2, ERK1/2, and mTOR, during heat shock (Fig. 5b)[48–50], and its phosphorylation was found to directly promote PIC formation via interaction with the SGO2-Pol II complex in mouse cells[21]. In contrast, HSF1-S419 was phosphorylated by PLK1, a central regulator of cell division that controls a variety of mitotic processes[70], but not by the other kinases described above (Fig. 5b)[18]. This phosphorylation triggered the recruitment of the TRRAP-TIP60 complex and TRIM33, which established an active chromatin state (Fig. 5f, g). Therefore, S419 is phosphorylated by a unique signaling pathway, and the mechanism by which its phosphorylation enhances transcriptional activity appears to differ from that mediated by S326 phosphorylation. This is why the mutation of both S326 and S419 markedly inhibited the HSF1 transcriptional activity upon heat shock (Supplementary Fig. 5b). Furthermore, the mutation of both S419 and S326 also had a profound impact on melanoma cell proliferation (Fig. 7). Our observations suggest that the phosphorylation-mediated HSF1 complexes including the TRRAP–TIP60 complex and the TRIM33–TRIM24 proteins are therapeutic targets for the treatment of melanoma patients.

## Methods

### Plasmid and adenovirus vectors

To generate expression vectors for wild-type hHSF1 tagged with hemagglutinin (HA) at the C-terminus (hHSF1-HA) and hHSF1-S419 phosphorylation-site mutants, including hHSF1-S419A-HA, hHSF1-S419G-HA, and hHSF1-S419V-HA, cDNA fragments were generated by PCR-mediated site-directed mutagenesis and inserted into pShuttle-CMV vectors (Stratagene) at the KpnI/XhoI sites. Expression vectors for a series of hHSF1 phosphorylation site mutants were similarly generated[15]. To generate an expression vector for hTRIM33 tagged with HA at the N-terminus (HA-hTRIM33), a hTRIM33 cDNA fragment (flanked by the KpnI and XhoI sites) was amplified by RT-PCR using total RNA isolated from HeLa cells, and then inserted into pShuttle-CMV vector. An expression vector for HA-hTRIM33-C125A/C128A, which lacks E3 ubiquitin ligase activity, was similarly generated using PCR-mediated site-directed mutagenesis. To generate an expression vector for hTRRAP (isoform 2, accession number NP_003487) tagged with 3×FLAG peptide at the C-terminus, a hTRRAP cDNA fragment was amplified by RT-PCR using total RNA isolated from HeLa cells, and inserted into pShuttle-CMV vector (the ClaI site was generated between the NotI and XhoI sites) at the NotI/ClaI sites. An expression vector for the deletion mutant hTRRAP-Δ1801-1833-3×FLAG, which cannot interact with hHSF1, was generated using PCR-mediated site-directed mutagenesis. Sequences of the expression vectors were verified using a 3500 Genetic Analyzer (Applied Biosystems). Adenovirus expression vectors including Ad-hHSF1-S419A-HA, Ad-hHSF1-S419G-HA, Ad-hHSF1-S419V-HA, Ad-hHSF1-S326A-HA, Ad-hHSF1-S326A/S419A-HA, Ad-HA-hTRIM33, and Ad-HA-hTRIM33-C125A/C128A were generated in accordance with the manufacturer's instructions (Agilent Technologies). Due to the upper limit of the size of the adenovirus genome containing a large hTRRAP cDNA fragment (-11,500 bp) for efficient virus packaging, we used the plasmid vectors pShuttle-CMV-hTRRAP-3×FLAG and pShuttle-CMV-hTRRAP −Δ1801-1833-3×FLAG for the overexpression of FLAG-tagged wild-type hTRRAP and mutant hTRRAP. To generate adenovirus vectors expressing short hairpin RNAs against human HSF1, TRRAP, TIP60, p400, p300, CBP, GCN5, PCAF, TRIM24, TRIM33, PLK1, CSNK1A1, NEK7, RNF20, and RNF40, oligonucleotides containing each target sequence (Supplementary

Table 1) were annealed and inserted into pCR2.1-hU6 at the BamHI/HindIII sites, and then XhoI/HindIII fragments containing hU6-shRNA were inserted into pShuttle-CMV vector (Stratagene)[15]. Viral DNA was generated in accordance with the manufacturer's instructions for an AdEasy adenoviral vector system (Agilent Technologies). Viral DNAs were infected into HEK293 cells and the virus particles were enriched by CsCl gradient centrifugation and stored at −80 °C until used.

## Cell cultures and treatments

Human cervical carcinoma HeLa cells (ATCC, CCL-2), melanoma MeWo (ATCC, HTB-65) and MMAc (RIKEN BRC, RCB0808) cells, mammary carcinoma MCF7 cells (RIKEN BRC, RCB1904), osteosarcoma U2OS cells (ATCC, HTB-96), hepatocyte carcinoma HepG2 cells (RIKEN BRC, RCB1648), colon cancer HCT116 cells (RIKEN BRC, RCB2979), lung carinoma A549 cells (RIKEN BRC, RCB0098), HEK293 (ATCC, CRL-1573), and immortalized fibroblast OUMS-36T-3F (JCRB1006.3F) and keratinocyte HaCat cells were maintained at 37 °C in 5% CO2 in Dulbecco's modified Eagle's medium (DMEM) (Sigma-Aldrich) containing 10% fetal bovine serum (FBS) (Sigma-Aldrich). Human melanoma HMV-1 cells (Cell Resource Center for Biomedical Research, Tohoku University, TKG 0302), prostate adenocarcinoma LNCap cells (RIKEN BRC, RCB2144), histiocyte lymphoma U-937 cells (ATCC, CRL-1593.2), pancreatic cancer PANC-1 cells (RIKEN BRC, RCB2095), and myeloma U266BB1 cells (ATCC, TIB-196) were maintained in RPMI 1640 medium (Gibco) containing 10% FBS. Human kidney adenocarcinoma ACHN cells (ATCC, CRL-1611) and glioblastoma U-87MG cells (ATCC, HTB-14) were maintained in Eagle's MEM medium (Gibco) containing 10% FBS. Cells were treated with heat shock at 42 °C for the indicated periods, 5 mM L-azetidine-2-carboxylic acid (AZC; Tokyo Chemical Industry) for 6 h, 10 or 20 μM MG132 (Sigma-Aldrich) for 6 h, or 50 μM sodium arsenite (As; Sigma-Aldrich) for 6 h. Cells were also pretreated for 3 h before heat shock with inhibitors of kinase activity, including 100 nM BI6727 (PLK1 inhibitor, Toronto Research Chemicals), 10 nM BI2536 (PLK inhibitor, Cayman Chemical), 1 μM FR180204 (ERK1/2 inhibitor, Tokyo Chemical Industry), 20 nM AZD6244 (MEK1/2 inhibitor, ChemScene), and 30 nM rapamycin (mTOR inhibitor, LC Laboratories). Regarding the double thymidine block, cells were treated with 2 mM thymidine (Sigma-Aldrich) for 16 h, washed three times with PBS, released for 8 h into normal medium, and then blocked again (2 mM thymidine, 16 h). To achieve the KD of HSF1, TRRAP, TIP60, p300, TRIM24, or TRIM33, HeLa cells were washed twice with PBS and infected with Ad-sh-hHSF1-KD, Ad-sh-hTRRAP-KD (KD1 or KD2), Ad-sh-hTIP60-KD, Ad-sh-hp300-KD, Ad-sh-hTRIM24-KD, or Ad-sh-hTRIM33-KD ($1 \times 10^7$ pfu/ml) in serum-free DMED for 2 h and maintained in normal medium for 70 h. To replace endogenous HSF1 with exogenous hHSF1 or its mutants, HeLa or OUMS-36T-3F cells were infected with Ad-sh-hHSF1-KD ($1 \times 10^7$ pfu/ml) for 2 h and maintained in normal medium for 22 h. Cells were then infected with Ad-hHSF1-HA, Ad-hHSF1-S419A-HA, Ad-hHSF1-S419G-HA, or Ad-hHSF1-S419V-HA ($2 \times 10^6$ pfu/ml) for 2 h and maintained with normal medium for a further 46 h. The replacement of endogenous TRIM33 with its mutant was performed in the same manner. To replace endogenous TRRAP, cells were infected with Ad-sh-hTRRAP-KD1 ($1 \times 10^7$ pfu/ml) for 2 h and maintained in normal medium for 22 h. Cells were then transfected for 48 h with pShuttle-CMV-hTRRAP-3 × FLAG or pShuttle-CMV-hTRRAP -Δ1801-1833-3 × FLAG by using Lipofectamine 3000 reagent (Invitrogen).

## ChIP assay

The chromatin immunoprecipitation (ChIP) assay was performed using a kit in accordance with the manufacturer's instructions (EMD Millipore). The following antibodies were used: antibodies for HSF1 (Millipore Sigma, ABE1044, 2 μl), hTRRAP (anti-hTRRAP-2, Nakai lab, 2 μl), Pol II (Millipore Sigma, 05-623, 2 μl), histone H3 (Abcam, ab1791, 2 μl), histone H3ac (acetylated at the N-terminus) (Millipore Sigma, 06-

599, 2 μl), histone H4 (Abcam, ab7311, 2 μl), histone H4ac (pan-acetyl) (Active Motif, 39925, 2 μl), TIP60 (Santa Cruz, sc-5725, 10 μl), p300 (Santa Cruz, C-20, 10 μl), H3K9ac (Millipore Sigma, 07-352, 2 μl), H3K18ac (Abcam, ab1191, 2 μl), H3K23ac (Abcam, ab177275, 2 μl), H3K27ac (Abcam, ab4729, 2 μl), H4K16ac (Abcam, ab109463, 2 μl), hTRIM24 (anti-hTRIM24-3, Nakai lab, 2 μl), hTRIM33 (anti-hTRIM33-2, Nakai lab, 2 μl), H2B (Abcam, ab1790, 2 μl), ubiquityl-histone H2B (Lys120) (Cell Signaling, 5546, 4 μl), H3K4me3 (Abcam, ab213224, 2 μl), and PLK1 (anti-PLK1-1, Nakai lab, 2 μl). Real-time qPCR of ChIP-enriched DNAs in *HSP72* (*HSPA1A*), *HSP27* (*HSPB1*), *HSP40* (*DNAJB1*), and *HSP110* (*HSPH1*) loci was performed using the primers listed in Supplementary Table 2[15]. Percentage input was assessed by comparing the cycle threshold value of each sample to a standard curve generated from a 5-point serial dilution of genomic input, and compensated by values obtained using normal IgG. IgG-negative control immunoprecipitations for all sites yielded <0.05% input. All reactions were performed in triplicate with samples derived from three experiments.

## Assessment of mRNA

Total RNA was extracted from HeLa cells using Trizol (Invitrogen), and first-strand cDNA was synthesized using PrimeScript II reverse transcriptase and oligo (dT)20 in accordance with the manufacturer's instructions (TAKARA). Real-time quantitative PCR (qPCR) for human HSP72 mRNA was performed using StepOnePlus (Applied Biosystems) with EagleTaq Master Mix with ROX (Roche) in accordance with the manufacturer's instructions, and relative quantities were normalized against GAPDH mRNA levels. The qPCR for human HSP110, HSP90α, HSP40, and HSP27 mRNAs was performed with Power SYBR Green PCR Master Mix (Applied Biosystems) and relative quantities were normalized against β-actin mRNA levels. Primers used for RT-qPCR reactions are listed in Supplementary Tables 3 and 4. All reactions were performed in triplicate with samples derived from three experiments.

## Western blotting

Cells were lysed on ice for 10 min in NP-40 lysis buffer [1.0% NP-40, 150 mM NaCl, 50 mM Tris-HCl (pH 8.0), and protease inhibitor cocktail (1 μg/ml leupeptin, 1 μg/ml pepstatin, and 1 mM phenylmethylsufonyl fluoride)] or RIPA lysis buffer [1.0% NP-40, 150 mM NaCl, 50 mM Tris-HCl (pH 8.0), 0.5% sodium deoxycholate, 0.1% SDS, and protease inhibitor cocktail]. After centrifugation at $16,000 \times g$ for 10 min, aliquots of the supernatant were subjected to SDS-PAGE. After transferal to a nitrocellulose membrane using Trans-Blot transfer cell (Bio-Rad), the membrane was blocked with 5% non-fat milk/PBS at room temperature (RT) for 1 h. The membrane was incubated with primary antibodies diluted in 2% milk/PBS at RT for 1 h or at 4 °C overnight. The following antibodies were used: rabbit antibodies for mHSF1 (anti-mHSF1n, Nakai lab, 1/1000), hTRRAP (anti-hTRRAP-2, Nakai lab, 1/1000), hTRIM24 (anti-hTRIM24-3, Nakai lab, 1/1000), hTRIM33 (anti-hTRIM33-2, Nakai lab, 1/1000), hPLK1 (anti-hPLK1-1, Nakai lab, 1/1000), GST (anti-GST, Nakai lab, 1/1000), mHSP110 (anti-mHSP110a, Nakai lab, 1/1000), hHSP90 (anti-hHSP90d, Nakai lab, 1/1000), hHSP40 (anti-hHSP40a, Nakai lab, 1/1000)[15], p300 (Santa Cruz, sc-585, 1/1000), p400 (Novus Biologicals, NB200-210, 1/1000), H2B (Abcam, ab1790, 1/1000), ubiquityl-histone H2B (Lys120) (Cell Signaling, 5546, 1/1000), RNF40 (GeneTex, GTX115295, 1/1000), histone H3 (Abcam, ab1791, 1/1000), H3 (acetyl K18) (Abcam, ab1191, 1/1000), Lamin B1 (Abcam, Ab16048, 1/1000), HSF1 (phosphor S326) (Abcam, ab115702, 1/1000) and HSF1 (phosphoS419) (anti-HSF1 phospho-S419b, Nakai lab, 1/1000), mouse antibodies for PLK1 (Santa Cruz, sc-17783, 1/1000), HSP72 (Santa Cruz, sc-24, 1/1000), ubiquitin (Santa Cruz, sc-8017, 1/1000), CSNK1A1 (Santa Cruz, sc-74582, 1/1000), NEK7 (Santa Cruz, sc-393539, 1/1000), RNF20 (Santa Cruz, sc-517358, 1/1000), β-actin (Milipore Sigma, A5441, 1/1000), and GFP (Nacalai Tesque, GF200, 1/1000), rat antibody for HA (Roche, ROAHAHA, 1/1000), and goat

antibody for TIP60 (Santa Cruz, N-17, 1/200). We generated rabbit antisera against hTRRAP (anti-hTRRAP-2) and hPLK1 (anti-hPLK1-1) by immunizing rabbits with bacterially expressed recombinant GST-hTRRAP (amino acids 3,467–3830) and GST-hPLK1 (amino acids 261–412), respectively. We also generated rabbit antisera against hTRIM24 (anti-hTRIM24-3) and hTRIM33 (anti-hTRIM33-2) by immunizing with GST-hTRIM24 (amino acids 628-828) and GST-hTRIM33 (amino acids 650-888), respectively. The membrane was washed three times with PBS for 5 min each, followed by incubation with horseradish peroxidase (HRP)-conjugated secondary antibodies (goat anti-rabbit IgG, MP Biomedicals, 55689, 1/2000; goat anti-mouse IgG, Jackson, 115-035-003, 1/1000; rabbit anti-goat IgG, MP Biomedicals, 55363, 1/1000; goat anti-rat IgG, Jackson, 112-035-003, 1/1000) in 2% milk/PBS at RT for 1 h. The membrane was washed three times with PBS containing 0.1% Tween 20 (PBS-T), and chemiluminescent signals from Amersham ECL detection reagents (GE Healthcare) were captured on an X-ray film (Super RX, Fujifilm). Apparent molecular weights on SDS-PAGE gels and Western blots were estimated using Precision Plus Protein Dual Color Standards (Bio-Rad, #1610374). Uncropped and unprocessed scans of all blots are provided in the Source Data file.

### ChIP-MS procedures

HeLa cells cultured in a 10 cm dish (Corning Inc., NY, USA) were treated or untreated with heat shock at 42 °C for 5, 30, and 60 min, and were then fixed in medium containing 1% formaldehyde at 37 °C for 10 min. Cells were washed twice with cold PBS, and suspended in 1.5 ml cell lysis buffer [LB1; 20 mM Tris-HCl (pH 7.5), 10 mM NaCl, 1 mM EDTA, 0.2% NP-40, and protease inhibitor cocktail] on ice for 10 min. Nuclei were pelleted by centrifugation at 1,500 × g for 5 min and suspended in 1.5 ml nuclei washing buffer [LB2; 20 mM Tris-HCl (pH 8.0), 200 mM NaCl, 1 mM EDTA, 0.5 mM EGTA, and protease inhibitor cocktail] on ice for 10 min. Nuclei were pelleted again and suspended in 1 ml sonication buffer [LB3; 20 mM Tris-HCl (pH 8.0), 150 mM NaCl, 1 mM EDTA, 0.5 mM EGTA, 1% Triton X-100, 0.1% sodium deoxycholate, 0.1% SDS, and protease inhibitor cocktail] at RT for 10 min, and then pelleted and resuspended in 400 μl sonication buffer on ice for 10 min. Resuspended chromatin was sonicated with the Branson Sonifier 450 (BRANSON Ultrasonics) into fragmented DNA of approximately 200-300 bp and centrifuged at 16,000 × g for 5 min. Sonicated chromatin in the supernatant was transferred into new tubes.

Fifty microliters of Invitrogen Dynabeads Protein A (Fisher Scientific) was washed twice with 5 mg/ml BSA in PBS, preincubated with 2 μl of rabbit immune serum for mHSF1 (anti-mHSF1j, Nakai lab)[14] or preimmune serum at 4 °C for 3 h, and then suspended in 100 μl of sonication buffer. These antibody-binding beads were incubated at 4 °C overnight with the sonicated chromatin as described above. Magnetic beads were washed 5 times with RIPA wash buffer [50 mM HEPES-KOH (pH 7.4), 0.25 M LiCl, 1 mM EDTA, 0.5% sodium deoxycholate, and 1% NP-40] and once in TE50 buffer [50 mM Tris-HCl (pH 8.0), 10 mM EDTA].

To elute chromatin proteins from the magnetic beads, the beads were then suspended in 40 μl of 1× Laemmli sample buffer [2% SDS, 60 mM Tris-HCl (pH 6.8), 100 mM DTT, 10% glycerol, and 0.001% bromophenol blue] and incubated at 95 °C for 30 min. After centrifugation at 16,000 × g for 5 min, 4 μl of the protein sample in the supernatant was loaded for 8% SDS-polyacrylamide gel electrophoresis (SDS-PAGE), and proteins on gels were visualized with a silver stain kit (Nacalai Tesque, Japan). Precision Plus Protein Dual-Color Standards (Bio-Rad) was used to estimate protein sizes. The other 36 μl of the protein sample was loaded for 12% SDS-PAGE to prepare gel slices for mass spectrometry (MS)[21]. Briefly, gel bands were cut-out and subjected to in-gel digestion with trypsin. The resulting peptides were dissolved in a solution containing 0.1% trifluoroacetic acid and 2% acetonitrile and analyzed by an LTQ Orbitrap Velos Pro mass spectrometer (Thermo Fisher Scientific, Waltham, MA) coupled with a nanoLC instrument (Advance, Michrom BioResources, Auburn, CA) and HTC-PAL autosampler (CTC Analytics, Zwingen, Switzerland). Peptide separation was performed with an in-house pulled fused silica capillary (internal diameter, 0.1 mm; length, 15 cm; tip internal diameter, 0.05 mm) packed with 3-μm C18 L-column (Chemicals Evaluation and Research Institute, Japan). The mobile phases consisted of 0.1% formic acid and 100% acetonitrile. Peptides were eluted with a gradient of 5–35% acetonitrile for 40 min at a flow rate of 200 nl/min. Collision-induced dissociation (CID) spectra were automatically acquired in the data-dependent scan mode with the dynamic exclusion option. Full MS spectra were obtained with Orbitrap in the mass/charge ($m/z$) range of 300–2000 with a resolution of 60,000 at $m/z$ 400. The 12 most intense precursor ions for in the full MS spectra were selected for subsequent ion-trap MS/MS analysis with the automated gain control (AGC) mode. The AGC mode was set to $1.00 \times 10^6$ for full MS and $1.00 \times 10^4$ for CID MS/MS. Normalized collision energy values were set to 35%. The lock mass function was activated to minimize mass errores during analyzes. Identified peptide numbers estimated with Mascot (MatrixScience) and LFQ intensity values calculated by MaxQuant software (https://www.maxquant.org) in HSF1 ChIP preparations were normalized by those in IgG ChIP preparations, and differences in the values in control and heat-shocked cells are calculated.

### Detection of phosphorylated proteins

To generate antibody specific for phosphorylated HSF1 at Ser419 (anti-HSF1 phospho-S419b), a phospho-S419-containing peptide (amino acid residues 416–428) flanked by a cysteine residue [DLF(pS)PSVTVPDMSC], was synthesized and used for immunization of rabbits after coupling to a carrier protein through the cysteine residue (Cosmo Bio Co., Otaru, Hokkaido, Japan). A specific antibody was purified using the phospho-peptide column, and then non-specific binding was further removed using the unphosphorylated peptide column. To identify HSF1-S419 phosphorylation, HeLa cells were treated with heat shock at 42 °C for the indicated periods, and then lysed in NP-40 lysis buffer containing 0.4 mM Na₃VO₄. After centrifugation at 16,000 × g for 10 min, aliquots of the supernatant containing 5 mg of proteins were incubated with 3 μl of rat anti-HSF1 monoclonal antibody (Novus Biologicals, 10H4) at 4 °C for 3 h, and then mixed with 50 μl protein G-Sepharose beads (GE Healthcare) by rotating at 4 °C for 2 h. Beads were washed five times with NP-40 lysis buffer and 50 μl of 2 × Laemmli sample buffer was then added. Alternatively, washed beads were incubated at 30 °C for 30 min in 50 μl of 1 × NEBuffer Pack for Protein MetalloPhosphatases containing lambda protein phosphatase (400 units, New England Biolabs), and sample buffer were then added. Proteins in the sample buffer were incubated at 95 °C for 5 min and subjected to SDS-PAGE. After transferal to a nitrocellulose membrane using Trans-Blot transfer cell (Bio-Rad), the membrane was blocked with 5% BSA/Tris-buffered serine containing 0.1% Tween 20 (TBS-T) at RT for 1 h. It was incubated with anti-HSF1 phospho-S419b antibody (Nakai lab, 1/500) in 5% BSA/TBS-T at 4 °C overnight. The membrane was washed with TBS-T three times for 5 min each, followed by incubation with HRP-conjugated goat anti-rabbit IgG (Cappel 55689, 1/2000) in 5% BSA/TBS-T at RT for 1 h. The membrane was washed three times with TBS-T, and chemiluminescent signals from Amersham ECL detection reagents (GE Healthcare) were captured on an X-ray film (Super RX, Fujifilm). To examine HSF1-S326 phosphorylation, total proteins in NP-40 lysis buffer were subjected to SDS-PAGE, transferred to a nitrocellulose membrane, and blotted with rabbit polyclonal antibody for HSF1 phospho-S326 (Abcam, ab115702, 1/1000) as described above.

## Identification of TRRAP-interacting proteins

To establish HeLa cells overexpressing hTRRAP-3×FLAG, cells were co-transfected with pShuttle-CMV-hTRRAP-3×FLAG and pcDNA3.1-Neo (Invitrogen) by using Lipofectamine 3000 (Invitrogen), and stable transformants were isolated in medium containing 1.0 mg/ml neomycin (Nacalai Tesque Inc., Kyoto, Japan). HeLa-hTRRAP-3×FLAG cells (clone 14) (two 10 cm dishes per each sample) were untreated or treated with heat shock at 42 °C for 10 or 30 min, and nuclear extracts were prepared and then diluted to generate extracts containing 150 mM NaCl and 1% NP-40 as described in the co-immunoprecipitation procedure. These nuclear extracts were mixed with 60 µl of anti-FLAG M2 affinity gels (Sigma-Aldrich) by rotating at 4 °C for 3 h. Affinity gels were washed three times with TBS, and were rotated at 4 °C for 30 min with 40 µl of the FLAG peptide (1 µg/µl) (Sigma-Aldrich, F3290). After centrifugation at 9500 × $g$ for 30 s, supernatants were moved to new tubes, 40 µl of 2 × Laemmli sample buffer was added, and supernatants were incubated at 95 °C for 5 min. Aliquots (10 µl each) were loaded on 8% SDS-PAGE, and proteins were visualized with a silver stain kit (Nacalai Tesque Inc., Kyoto, Japan) or subjected to immunoblotting using anti-FLAG antibody (Sigma-Aldrich, F3165, clone M2, 1/1000). The other 60 µl protein of samples was loaded on 12% SDS-PAGE for the preparation of gel slices for MS as described above[21]. Identified peptide numbers estimated with Mascot (MatrixScience) and LFQ intensity values calculated by MaxQuant software (https://www.maxquant.org) in co-immunoprecipitated complexes were normalized by those in co-immunoprecipitated complexes from HeLa cells.

## GST pull-down assay

To generate bacterial expression vectors for hTRRAP deletion mutants fused to GST, we initially generated pGEX-2T-MCS1 vector by inserting a DNA fragment containing XhoI restriction sites into pGEX-2T vector at the BamHI/EcoRI sites (GE Healthcare). The cDNA fragments encoding deleted hTRRAP proteins were generated by PCR using pShuttle-CMV-hTRRAP-3×FLAG plasmid as a template, and were then inserted into pGEX-2T-MCS1 vector at the BamHI/EcoRI, BamHI/XhoI, or XhoI/EcoRI sites. Recombinant GST-hTRRAP deletion mutants were expressed in *Escherichia coli* strain DH5α by incubating with 0.4 mM isopropyl β-D-1- thiogalactopyranoside at 37 °C for 3 h. Bacterial cultures were centrifuged at 4400 × $g$ for 5 min, and pellets were suspended in STE buffer [10 mM Tris-HCl (pH 8.0), 150 mM NaCl, 1 mM EDTA, 5 mM DTT, and 1.5% Sarkosyl) and then sonicated. After addition of Triton X-100 to a final concentration of 2%, bacterial extracts were prepared by removing pellets after centrifugation at 174,000 × $g$ for 10 min, and GST-fused proteins were purified using Glutathione Sepharose 4B (GE Healthcare). A mixture of each purified GST-fused protein (1 µg) and an extract (0.5 mg) of heat-shocked (42 °C for 30 min) HeLa cells in NP-40 lysis buffer was rotated at 4 °C for 1 h, and was then incubated with 20 µl of Glutathione-Sepharose beads at 4 °C for 1 h. After the beads had been washed five times with NP-40 lysis buffer, bound proteins were resuspended in 40 µl of 2 × Laemmli sample buffer and subjected to immunoblotting using antibodies for HSF1 (anti-mHSF1n, Nakai lab, 1/1000) and GST (anti-GST, Nakai lab, 1/1000).

## ChIP-seq and data analysis

HeLa cells treated or untreated with heat shock at 42 °C for 30 min were fixed in 1% formaldehyde-containing medium at 37 °C for 10 min. Cells were washed twice with cold PBS, and suspended in 1.5 ml cell lysis buffer [LB1; 20 mM Tris-HCl (pH 7.5), 10 mM NaCl, 1 mM EDTA, 0.2% NP-40, protease inhibitor cocktail] on ice for 10 min[22]. Nuclei were pelleted and suspended in 1.5 ml nuclei washing buffer [LB2; 20 mM Tris-HCl (pH 8.0), 200 mM NaCl, 1 mM EDTA, 0.5 mM EGTA, protease inhibitor cocktail] on ice for 10 min. Nuclei were pelleted and suspended in 1 ml sonication buffer [LB3; 20 mM Tris-HCl (pH 8.0),

150 mM NaCl, 1 mM EDTA, 0.5 mM EGTA, 1% Triton X-100, 0.1% Na-Deoxycholate, 0.1% SDS, protease inhibitor cocktail] at room temperature for 10 min, and then pelleted and resuspended in 400 µl sonication buffer on ice for 10 min. Resuspended chromatin was sonicated with the Sonifier 450 (Branson Ultrasonics) into fragmented DNA of ~200–300 bp, centrifuged at 16,000 × $g$ for 5 min, and transferred into new tubes. Fifty microliters of Dynabeads Protein A (Invitrogen) was washed twice with 5 mg/ml BSA in PBS, preincubated with 2 µg of an appropriate antibody at 4 °C for 3 h, and suspended in 100 µl sonication buffer. Antibodies used were rabbit immune serum for mHSF1 (anti-mHSF1j, Nakai lab), hTRRAP (anti-hTRRAP-2, Nakai lab), hTRIM24 (anti-hTRIM24-3, Nakai lab), or hTRIM33 (anti-hTRIM33-2, Nakai lab). These beads were incubated at 4 °C overnight with sonicated chromatin, and were washed 5 times with RIPA wash buffer [50 mM HEPES-KOH (pH 7.4), 0.25 M LiCl, 1 mM EDTA, 0.5% Na-Deoxycholate, 1% NP-40] and once with TE50 buffer [50 mM Tris-HCl, 10 mM EDTA]. To elute chromatin from the magnetic beads, the beads were then suspended in 200 µl of EB buffer [TE50 buffer containing 1% SDS] and incubated at 65 °C for 20 min. Eluted chromatin was de-crosslinked at 65 °C for 6 h, and was treated with RNase A at 50 °C for 1 h and then with proteinase K at 50 °C overnight. DNA was extracted with phenol-chloroform, followed by ethanol precipitation, and was then purified using QIAquick PCR Purification Kit (QIAGEN). The ChIP-seq libraries were prepared using the NEBNext Ultra II DNA Library Prep Kit for Illumina (New England Biolabs), and were run on the HiSeq2000 sequencer (Illumina) to generate single-end 65-bp reads.

Sequenced reads obtained by performing ChIP-seq (see Supplementary Materials and Methods) were mapped to the human genome (UCSC hg38) using Bowtie[71] version 1.1.2 with "-n2 -m1" option, which allows two mismatches in the first 28 bases per read and outputs the uniquely mapped reads. Redundantly mapped reads (reads starting exactly at the same 5′-sequence ends) were filtered out for further analysis. For peak-calling and data-visualization, we used DROMPA3[72] version 3.7.2 with default parameter set, which normalizes the read distribution with the total number of mapped reads and identifies the regions that satisfy following criteria. $P$ value of ChIP read enrichment, $p < 1 \times 10^{-4}$, and $P$ value of fold enrichment (ChIP/Input), $p < 1 \times 10^{-4}$.

## Co-immunoprecipitation

HeLa cells cultured in two 10 cm dishes were harvested, washed twice with ice-cold PBS and once with 500 µl buffer A [10 mM HEPES-KOH (pH 7.9), 10 mM KCl, 1.5 mM $MgCl_2$, 0.5 mM DTT, and protease inhibitor cocktail), and then resuspended in 300 µl of buffer A. After incubation for 15 min on ice, the cells were homogenized using a tight fitting Dounce homogenizer (Weaton type A, 40 strokes), and centrifuged at 500 × $g$ at 4 °C for 5 min. The supernatant was diluted with an equal volume (300 µl) of 2 × NP-40 lysis buffer. On the other hand, the pellet was suspended in 300 µl of buffer C [20 mM HEPES-KOH (pH 7.9), 25% glycerol, 0.42 M NaCl, 1.5 mM $MgCl_2$, 0.2 mM EDTA, 0.5 mM DTT, and protease inhibitor cocktail][73] and sonicated using Sonifier 450 (Branson Ultrasonics). After centrifugation at 16,000 × $g$ for 10 min, the supernatants were diluted with 1.8 volume (540 µl) of dilution buffer [50 mM Tris-HCl (pH 8.0) and 1.35% NP-40] to generate nuclear extracts containing 150 mM NaCl and 1% NP-40. Cytoplasmic and nuclear extracts isolated from an equal number of cells were incubated with 2 µl of rabbit non-immune serum or immune serum for hTRRAP (anti-hTRRAP-2, Nakai lab), PLK1 (anti-hPLK1-1, Nakai lab) or mHSF1 (anti-mHSF1n, Nakai lab) on ice for 3 h, and mixed with 20 µl of protein A-Sepharose beads (GE Healthcare) by rotating at 4 °C for 1 h. Beads were washed five times with NP-40 lysis buffer, followed by addition of 20 µl of 2 × Laemmli sample buffer. Proteins in the sample buffer were incubated at 95 °C for 5 min and subjected to immunoblotting using appropriate antibodies. To detect interactions of HSF1 with histone H3 and H3K18ac, HeLa cells were harvested, washed twice with ice-cold PBS, and incubated on ice for 30 min with 500 µl PBS

containing the chemical crosslinking reagent DSP [dithiobis(succinimidyl propionate)] (Thermo Fisher Scientific) at a concentration of 1 mM. The cross-linking reaction was stopped by addition of stop solution [10 mM Tris-HCl (pH 7.4) and 570 mM glycine] at RT for 5 min. After the cells were washed three times with PBS, nuclear extracts were prepared and subjected to co-immunoprecipitation analysis.

## Immunofluorescence

Hela cells were grown on glass coverslips (Matsunami Glass Ind., Osaka, Japan) at 37 °C for 16 h in 5% $CO_2$, and were transfected for 48 h with pShuttle-CMV-hHSF1-HA, pShuttle-CMV-hHSF1-S419A-HA, or pShuttle-CMV-hHSF1-S419G-HA using Lipofectamine 3000 reagent (Invitrogen). Cells were untreated or treated with heat shock at 42 °C for 30 min, and then fixed with 4% paraformaldehyde in PBS at RT for 10 min, and then washed three times with PBS for 5 min each. Cells were permeabilized with 0.2% Triton X-100/PBS at RT for 10 min and blocked with 5% non-fat milk/PBS at RT for 10 min. After washing with PBS once, coverslips were incubated with rat antibody for HA (Roche, 3F10) (1/1000 dilution in 2% milk/PBS) at RT for 1 h, and washed three times with PBS. They were then incubated with Alexa Fluor 546-conjugated goat anti rat IgG (Thermo Fisher Scientific, A-11081, 1/200 dilution) at RT for 1 h. Coverslips were washed three times with PBS for 5 min each, and then mounted in VECTASHIELD with 4′−6-diamino-2-phenylindole (DAPI) mounting medium (Vector Laboratories). Fluorescence images were taken using LSM510 META confocal microscope (Carl Zeiss).

## Fractionation of the nucleoplasm and chromatin

HeLa cells were harvested, washed in ice-cold PBS, and resuspended in five packed cell pellet volumes of buffer A [10 mM HEPES-KOH (pH 7.9), 10 mM KCl, 1.5 mM $MgCl_2$, 0.5 mM DTT, and protease inhibitor cocktail][73]. After incubation for 15 min on ice, cells were homogenized using a tight fitting Dounce homogenizer (Weaton type A, 40 strokes), and centrifuged at $500 \times g$ at 4 °C for 10 min. To prepare nucleoplasmic and chromatin-bound fractions, the nuclear pellets were homogenized in a buffer containing 0.5% NP-40, 10 mM HEPES-KOH (pH 7.9), 75 mM KCl, 1.5 mM $MgCl_2$, and protease inhibitor cocktail, and centrifuged at $16,000 \times g$ at 4 °C for 10 min. Supernatants (nucleoplasmic fractions) were collected, and the pellets were homogenized in buffer C [20 mM HEPES-KOH (pH 7.9), 25% glycerol, 0.42 M NaCl, 1.5 mM $MgCl_2$, 0.2 mM EDTA, 0.5 mM DTT, and protease inhibitor cocktail], sonicated using Sonifer 450 (Branson Ultrasonics), and centrifuged at $16,000 \times g$ for 10 min to obtain chromatin-bound fractions. Aliquots extracted from equal numbers of cells were subjected to immunoblotting using antibodies for TRIM33 (anti-hTRIM33-2, Nakai lab, 1/1000), TRIM24 (anti-hTRIM24-3, Nakai lab, 1/1000), PLK1 (anti-hPLK1-1, Nakai lab, 1/1000), HSF1 (anti-mHSF1n, Nakai lab, 1/1000), histone H3 (Abcam, ab1791, 1/1000), and β-actin (Milipore Sigma, A5441, 1/1000).

## Assessment of proteostasis

To generate the expression vector pact-luc, a cDNA fragment coding for the human β-actin promoter (−1521 to +860) was amplified by PCR and inserted into the ptk-galp3-luc vector at the Hind III/BglII sites[74]. HeLa cells were co-transfected with pact-luc and pcDNA3.1-Neo (Invitrogen) using Lipofectamine 3000 (Invitrogen), and stable transformants (HeLa-luc) were isolated in medium containing 1.5 mg/ml neomycin (Nacalai Tesque Inc. Kyoto, Japan). HSF1-KD, TRRAP-KD or scrambled RNA-treated HeLa-luc cells (clone 23) cultured in 6-cm dishes were untreated or treated with heat shock at 42 °C for 2 h to denature the luciferase sensor protein, and were then recovered at 37 °C for the indicated periods. To measure luminescence, cells were washed three times with PBS, lysed in 350 µl Triton/glycylglycine lysis buffer [1% Triton X-100, 25 mM glycylglycine (pH 7.8), 15 mM $MgSO_4$, 4 mM EGTA, and 1 mM DTT], and centrifuged at $16,000 \times g$ at 4 °C for

5 min. The supernatant (10 µl) was mixed with 220 µl of luciferase assay buffer [25 mM glycylglycine (pH 7.8), 15 mM potassium phosphate, pH 7.8, 15 mM $MgSO_4$, 4 mM EGTA, 2 mM ATP, and 1 mM DTT]. After injection of 100 µl D-luciferin (0.2 mM) (Sigma) into a sample, light output at 25 °C for 10 s was measured using AB-2200 Luminescencer (ATTO Co., Tokyo). Luciferase activity values were normalized to the value of control cells maintained at 37 °C. To estimate the accumulation of insoluble ubiquitylated proteins during heat shock, HeLa cells, in which endogenous HSF1 was replaced with GFP, wild-type hHSF1-HA, hHSF1-S419A-HA, or hHSF1-S419G-HA, were heat-shocked at 45 °C for 3 h. Cells were then lysed in NP-40 lysis buffer, and centrifuged at $16,000 \times g$ at 4 °C for 10 min. Insoluble ubiquitylated proteins were examined by immunoblotting using anti-ubiquitin antibody (P4D1, Santa Cruz, 1/1000). Chemiluminescent signals from ECL detection reagents were captured on an X-ray film, scanned, and quantified using ImageJ (https://imagej.net).

## Growth of cancer cells

Various types of human cancer cells and normal cells were maintained at 37 °C in 5% $CO_2$ in DMEM (Sigma-Aldrich), Eagle's MEM (Sigma-Aldrich), or RPMI1640 (Gibco) containing 10% FBS (Sigma-Aldrich). To replace endogenous HSF1 with its phosphorylation site mutants, cells in 10-cm dishes were infected with Ad-sh-hHSF1-KD or Ad-sh-SCR ($0.5–1 \times 10^7$ pfu/ml) for 2 h and maintained in normal medium for 22 h. Cells were then infected with Ad-hHSF1-HA, Ad-hHSF1-S419A-HA, Ad-hHSF1-S419G-HA, Ad-hHSF1-S326A-HA, Ad-hHSF1-S326A/S419A-HA, or Ad-GFP ($0.1–2 \times 10^6$ pfu/ml) for 2 h and maintained in normal medium for a further 46 h. These cells were divided into 35-mm dishes and maintained for the indicated periods, and the number of cells was counted until 144 h. Some cells were maintained in the presence of the MEK inhibitor trametinib at a concentration of 10 nM to 1 µM. Alternatively, extracts of these cells were subjected to immunoblotting using antibody for HSF1 or β-actin, and the results obtained showed that the levels of ectopically expressed hHSF1 mutants were similar to those endogenous HSF1. To examine the effects of HSF1 phosphorylation on tumor formation, MeWo cells were inoculated into 5-week-old male athymic nude mice (BALB/cSlc-nu/nu, SLC Inc., Japan). Mixtures of cells suspended in 100 µl PBS and 100 µl Matrigel (BD Biosciences) were subcutaneously injected into anesthetized mice with a 23-gauge needle. Tumor volumes were calculated at each time point after the inoculation using the following equation: tumor volume ($mm^3$) = (length × width$^2$)/2. Tumors were harvested from euthanized mice. Mice were killed when loss of ability to ambulate was observed or at the experiment endpoint with rapid and humane method of euthanasia. The maximum tumor size allowed by our ethics committee of 2000 $mm^3$ was not exceeded in these studies. All experimental protocols related to these mice were approved by the Committee for Ethics on Animal Experiments of Yamaguchi University Graduate School of Medicine.

## Statistical analysis

Data were analyzed using the Student's *t* test, two-way ANOVA, or one-way ANOVA, followed by Tukey-Kramer post hoc test (JMP Pro 16 software, SAS Institute Inc., Cary, NC, USA). Error bars represent the standard errors of the means for more than three independent experiments (*n* = number of independent experiments).

## Reporting summary

Further information on research design is available in the Nature Research Reporting Summary linked to this article.

# Data availability

The data that support this study are available from the corresponding author upon reasonable request. The ChIP-MS data and TRRAP-interacting proteins MS data generated during this study are available

at jPOSTrepo (Japan ProteOme STandard Repository) with PXD accession codes PXD031821 and PXD031822. The ChIP-seq data are available at NCBI BioProject with BioProject ID PRJNA811704. Source data are provided with this paper.

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

## Acknowledgements

We are grateful to Drs. Mizuho Oda, Emiko Koba, and Naoko Yokota for their expert technical assistance. We thank Drs. Hideyasu Matsuyama, Masahiro Abe, and Toshihiko Torigoe for reagents. This work was supported by JSPS KAKENHI grant numbers 21H02697 (to A.N.) and 19K06490 (to M.F.), the Uehara Memorial Foundation (to M.F.), the Takeda Science Foundation (to R.T.), the Cooperative Research Project Program of the Medical Institute of Bioregulation, Kyushu University (to A.N., M.M., K.I.N), and the Platform Project for Supporting Drug Discovery and Life Science Research (Basis for Supporting Innovative Drug Discovery and Life Science Research (BINDS)) from AMED under Grant Number JP21am0101105 (support number 2611) (to A.N., K.S.).

## Author contributions

Conceptualization, M.F., A.N.; Methodology, M.F., M.M., K.I.N., R.N., K.F., K.S.; Investigation, M.F., R.T., M.M., R.N.; Writing Original Draft, M.F., A.N.; Writing Review and Editing, M.F., R.T., M.O., M.M., R.N., A.N.; Funding Acquisition, M.F., R.T., A.N.; Supervision, A.N.

## Competing interests

The authors declare no competing interests.

## Additional information

**Peer review information** *Nature Communications* thanks Chengkai Dai and Other anonymous Reviewer(s) to the peer review of this work. Peer review Reports are available.

