## [Peer Review File · Nature Communications]

REVIEWER COMMENTS

Reviewer #1 (Remarks to the Author):

In this manuscript (#NCOMMS-22-12540), Dr. Fujimoto and colleagues discovered that during heat shock, PLK1 phosphorylates HSF1 at Ser419 on the promoters of some HSF1 transcriptional targets. Focusing on the HSP72 promoter, they showed that this Ser419 phosphorylation recruits the TRRAP-TIP60 acetyltransferase complex to modify histones. Subsequently, bromodomain-containing ubiquitin ligases TRIM24 and TRIM33 recognize these histone acetylation marks to mono-ubiquitinate H2B, thereby stabilizing the HSF1 transcription complex on the HSP72 promoter and promoting the HSP72 transcription. Apart from its involvement in the heat-shock response, this mechanism also plays an important role in cancer. Consistent with the key pro-oncogenic role of HSF1, this PLK1-mediated Ser419 phosphorylation is critical to the proteomic stability, survival, and growth of cancer cells in vitro. Importantly, this phosphorylation is also required for in vivo melanoma growth.

The authors conducted a comprehensive study that elucidates a fundamental mechanism underlying the canonical HSF1-mediated heat shock response, which has important implications in both stress and cancer biology. Large amounts of data are presented in this manuscript, which are of high quality and convincing. Collectively, the data support the main conclusions. In summary, the reviewer is enthusiastic about and recommend publication of the novel findings reported in this manuscript. Some minor questions and points should be clarified before publication.

1. In Figure 1D, only 28 ChIP-seq peaks overlap between HSF1 and TRRAP, especially after HS, despite over 3700 peaks bound by HSF1. If technical issues can be ruled out, this is intriguing. Does it suggest that the TRRAP-TIP60 acetyltransferase complex is only recruited to a small number of HSF1 target genes during HS? Then, is the expression of other HSF1 target genes mediated through different mechanisms? It would be helpful to discuss this point in the manuscript.

2. In Figure 2G, the S419A or S419G substitution resulted in an impaired HSF1 DNA binding, which suggests that the TIP60-mediated histone acetylation may stabilize the HSF1 DNA binding at target sites. Although TRRAP KD also impaired HSF1 DNA binding, the TIP60-independent functions of TRRAP cannot be excluded. It would be interesting to see if TIP60 KD also impair the HSF1 DNA binding.

3. In Supplementary Figure 3A and 3E, there are some interesting findings. There are more H3 and H4 deposited at the HSP72 promoter near HSEs following HSF1 KD. Do the authors have some explanations?

4. Regarding the statistical methods, in general wherever the data contain multiple groups (e.g. Figure 2G), one-way ANOVA is more appropriate than Student's t-test.

5. It is recommended to cite one reference (<https://pubmed.ncbi.nlm.nih.gov/26597576/>) in this manuscript.

6. On page 17 line 305, "cures" should be "cues".

7. On page 19 line 348, "both" should be deleted.

8. Throughout the manuscript, HSP70 should be defined as HSP72 or HSPA1A, as there are several HSP70 family members.

9. For clarity, different HSEs should be first defined in the legend of Figure 2G. d: distal; p: proximal; and inter: intergenic.

Reviewer #2 (Remarks to the Author):

This work by Fujimoto et al. addresses the question of how an open chromatin state, that is permissible for transcription factor binding and transcription initiation, is established. While this is a fundamental process in the biology of gene expression, little is known about the mechanism or the assemblage of proteins that drive this process for a specific transcription factor. Given the extensive experience and discoveries made previously in the Nakai laboratory on the Heat Shock Transcription Factor 1 (HSF1), the authors of this work approach this question by dissecting the mechanisms specific to HSF1, a stress-activated transcription factor. Their observations are likely to be relevant to HSF1 activation by acute stress and in response to oncogenic signaling, and will have more general applicability to other transcription factors.

In this work Fujimoto and colleagues have carefully elucidated the factors, post-translational modifications and biological importance of establishing open chromatin structure for HSF1.

The manuscript presents a large quantity of high quality, rigorous and compelling data from well-controlled experiments that strongly support the conclusions drawn by the authors and the model presented in Figure 7H. The experimental approaches are wide-ranging, incisive and very appropriate for deciphering the questions addressed by the authors. Overall, this work makes a very important contribution to the field of understanding stress-responsive gene expression.

Suggestions/Questions:

1. The data on the co-occupancy of HSF1 and TRRAP on 28 loci in heat shocked cells is convincing. Yet it provokes the question that of the hundreds of direct HSF1 target genes in response to heat shock, why is there only co-occupancy at 28 loci? Could the authors speculate or clarify this point in the manuscript?
2. In Figure 2A, the authors should describe what the color-coded regions in the model for the HSF1 primary structure are. Although the authors described the DHR domain in previous work (Nakai et al, 1997) what is the significance of this region since that is where Serine 419 appears to lie within? Given that the DHR is conserved in other HSF isoforms outside of HSF1 (see Nakai group previous publications) the lineup in Figure 2e seems to artificially focus on the DHR of HSF1 in different species, but couldn't this region also be functional for other HSF isoforms if the S419 is conserved? Please address this point.
3. In Figure 2C and related primary and supplementary figures showing P-S419 by Western blotting, could the authors point out exactly which band represents the P-S-419 phosphorylated HSF1? It is currently somewhat confusing. Also, in Figure 5a, why is there no heat shock induction of S419 phosphorylation over control levels at 60 minutes, as there is in Figure 2a? This seems inconsistent with the nice time course in Figure 2a.
4. The authors could discuss more about the significance of PLK-1 regulation during the cell cycle. What is the significance of this observation with respect to HSF1 regulation? Could it be related to decreased chromatin access during mitosis as reported by others? Cell cycle regulation of PLK-1 has already been established in the literature at both the protein and mRNA level.
5. The authors accurately state that a previous report showed that PLK-1 phosphorylates HSF1 at Serine 419 in vitro. However, this same reference (Kim et al. 2005, Journal of Biological Chemistry) also presented data to support (1) an interaction between HSF1 and PLK-1 in vivo, (2) that this interaction increases in response to heat shock stress, and (3) that PLK-1 function positively influences HSF1 nuclear accumulation, which could be related to the open chromatin phenomenon beautifully deciphered in the current manuscript. Although in the report by Kiim et al. the mechanisms underlying the role of PLK-1 were not elucidated, the authors could incorporate these previous observations into their manuscript. This would not detract from the novelty of their findings, but rather the prior findings would strongly

support the elegant mechanistic dissections that the current authors have done on the role of PLK-1 in the assembly of the HSF1 complex.

6. Page 15, line 269: the term “highly phosphorylated” seems to not adequately describe the observation made. The authors could more accurately describe what was observed in terms of the level of phosphorylation observed.

7. Page 17, line 305: “cures” should be “cues”

8. Page 15, line 275: The authors note that HSF1 KD is more inhibitory to cell proliferation in melanoma cells compared to other cancer cells. Could the authors speculate on whether this could be due to the levels of TRRAP, TRIM24, TRIM33, PLK-1 mediated HSF1 Ser419 phosphorylation or other aspects of the levels or function of the complex shown in Figure 7H? What do the authors think might underlie this differential sensitivity?

9. From a technical perspective, can the authors clarify whether the chromatin IP and proteomics studies were conducted on cells expressing endogenous levels of HSF1 or were the cells over-expressing HSF1? Please clarify in the text of the manuscript.

10. The authors don't have to address this, but it would be very interesting to investigate how the chromatin-opening complex on HSF1 might change between genes activated by an acute heat shock target and those activated by oncogenic signaling in cancer cells. Additionally, are the TRAPP, etc. proteins important for genes whose expression is repressed by HSF1?

Reviewer #3 (Remarks to the Author):

In this manuscript, Fujimoto et al. describe that HSF1 phosphorylation at S419 recruits the TRRAP-TIP60 acetyltransferase complex and thereby promotes an active chromatin state. S419 is phosphorylated after HS through the PLK1 kinase. HSF1 and a TIP60 dependent acetylation mark together recruit the TRIM33 ubiquitin ligase that promotes H2B ubiquitination. Together, this establishes an active

chromatin state conducive to transcription. The authors go on to show that HSF1 S419 phosphorylation is constitutively enhanced in melanoma cells and promotes their proliferation.

The topic is highly interesting and timely. The question of how sequence-specific transcription factors promote transcriptional activation is still unresolved, despite much work on it. This manuscript provides a novel mechanism at the molecular level that goes a long way towards addressing this question. As the identified phosphorylation mark is upregulated in tumor cells, it constitutes a potential target for anti-cancer therapies. The experiments are thoroughly conducted and the conclusions are well justified. The methodology is described in detail. I do not have any major comments for improvement.

Minor comments:

- Fig. 1e: The authors should spell out in the legend what an MA plot is.
- l. 861, l.898, 925: Does n refer to biological or technical replicates?

Point-By-Point Response to the reviewer's comments

RE: Nature Communications Manuscript NCOMMS-22-12540

**** Reviewer #1 ****

The authors conducted a comprehensive study that elucidates a fundamental mechanism underlying the canonical HSF1-mediated heat shock response, which has important implications in both stress and cancer biology. Large amounts of data are presented in this manuscript, which are of high quality and convincing. Collectively, the data support the main conclusions. In summary, the reviewer is enthusiastic about and recommend publication of the novel findings reported in this manuscript. Some minor questions and points should be clarified before publication.

Thank you for favorable and constructive comments. We addressed the reviewer's comments as described below.

1. In Figure 1D, only 28 ChIP-seq peaks overlap between HSF1 and TRRAP, especially after HS, despite over 3700 peaks bound by HSF1. If technical issues can be ruled out, this is intriguing. Does it suggest that the TRRAP-TIP60 acetyltransferase complex is only recruited to a small number of HSF1 target genes during HS? Then, is the expression of other HSF1 target genes

mediated through different mechanisms? It would be helpful to discuss this point in the manuscript.

Thank you for pointing it out. The TRRAP antibody (anti-hTRRAP-2, this study) generated lower signal-to-noise ratios using HeLa cells or HeLa cells overexpressing hTRRAP-3×FLAG in ChIP assay than that generated by HSF1 antibody (see Fig. 2g). Because of this technical issue, only genome sites abundantly occupied by TRRAP were detected in HeLa cells and the numbers of identified TRRAP binding peaks were limited using ChIP-seq in Fig. 1d. We modified the statement in the results section as below.

“Although the numbers of identified TRRAP binding peaks were limited, they co-occupied 28 sites in heat-shocked cells, including many sites within the gene promoters of major HSPs, co-chaperone (p23), and ubiquitin (UBB) (Fig. 1d-f and Supplementary Fig. 1e, f).” (P6-P7)

We also add the statement in the legend of Fig. 1d as below.

“Because the TRRAP antibody generates a low signal-to-noise ratio in ChIP assay, the limited numbers of TRRAP binding peaks were identified using ChIP-seq.” (P33-P34)

2. In Figure 2G, the S419A or S419G substitution resulted in an impaired HSF1 DNA binding, which suggests that the TIP60-mediated histone acetylation may stabilize the HSF1 DNA binding at target sites. Although TRRAP KD also impaired HSF1 DNA binding, the TIP60-independent functions of TRRAP cannot be excluded. It would be interesting to see if TIP60 KD also impair the HSF1 DNA binding.

Thank you for pointing it out. We now show that TIP60 KD (or p300 KD) moderately impaired the HSF1 binding to *HSP72* promoter in new Supplementary Fig. 3c. We added the statement in the result section as below.

“ChIP assay showed that TIP60 occupied *HSP72* promoter during heat shock, but did not occupy it when HSF1 was KD or substituted with hHSF1-S419 mutants (Fig. 3d). Conversely, TIP60 KD moderately impaired the HSF1 binding to the promoter (Supplementary Fig. 3c).” (P9)

TRRAP KD has a stronger effect on the HSF1 DNA binding than TIP60 KD, probably because TRRAP not only recruits TIP60 but also other factors including the ATP-dependent chromatin remodeling protein p400 (Fig. 3b, c).

3. In Supplementary Figure 3A and 3E, there are some interesting findings. There are more H3 and H4 deposited at the HSP72 promoter near HSEs following HSF1 KD. Do the authors have some explanations?

Thank you for pointing it out. We previously reported that the deposition of histone H3 and H2B at *HSP72* promoter is elevated upon HSF1 KD in mouse MEF cells, because HSF1 constitutively binds to the promoter and displaces histones from nucleosomes in complex with RPA, the histone chaperone FACT, and the BRG1-containing chromatin remodeling complex (Fujimoto et al., *Mol. Cell* 2012). In Supplementary Fig. 3A and 3E, we similarly show that the deposition of histone H3 and H4 at *HSP72* promoter is elevated upon HSF1 KD in human HeLa cells. Therefore, in the legend of Supplementary Fig. 3a, we now added an explanation of these findings as below.

“Occupancy of H3 and H4 was elevated upon HSF1 KD in control cells because HSF1 constitutively binds to *HSP72* promoter and displaces histones from nucleosomes in complex with RPA, the histone chaperone FACT, and the BRG1-containing chromatin remodeling complex (Fujimoto et al., *Mol. Cell* 2012).“ (Supplementary Information, P8)

4. Regarding the statistical methods, in general wherever the data contain multiple groups (e.g. Figure 2G), one-way ANOVA is more appropriate than Student’s t-test.

Thank you for pointing it out. Now, one-way ANOVA (followed by Tukey-Kramer test) was used in statistical test for most data contain multiple groups.

5. It is recommended to cite one reference (<https://pubmed.ncbi.nlm.nih.gov/26597576/>) in this manuscript.

We cited the above reference (reference number 52).

6. On page 17 line 305, “cures” should be “cues”.

We amended it.

7. On page 19 line 348, “both” should be deleted.

We amended it.

8. Throughout the manuscript, HSP70 should be defined as HSP72 or HSPA1A, as there are several HSP70 family members.

As the reviewer suggested, we defined human HSP70 (HSPA1A) as HSP72 in the text.

9. For clarity, different HSEs should be first defined in the legend of Figure 2G. d: distal; p: proximal; and inter: intergenic.

In the legend of Fig. 2g, we added definition of different HSEs as was suggested.

**** Reviewer #2 ****

The manuscript presents a large quantity of high quality, rigorous and compelling data from well-controlled experiments that strongly support the conclusions drawn by the authors and the model presented in Figure 7H. The experimental approaches are wide-ranging, incisive and very appropriate for deciphering the questions addressed by the authors. Overall, this work makes a very important contribution to the field of understanding stress-responsive gene expression.

Thank you for favorable and careful comments. We addressed the reviewer's concerns as described below.

Suggestions/Questions:

1. The data on the co-occupancy of HSF1 and TRRAP on 28 loci in heat shocked cells is convincing. Yet it provokes the question that of the hundreds of direct HSF1 target genes in response to heat shock, why is there only co-occupancy at 28 loci? Could the authors speculate or clarify this point in the manuscript?

Thank you for pointing it out. We cannot answer the question of whether a lot of HSF1 targets (other than 28 targets) were co-occupied by TRRAP or not at this moment. The TRRAP antibody (anti-hTRRAP-2, this study) generated lower signal-to-noise ratios using HeLa cells or HeLa cells overexpressing hTRRAP-3×FLAG in ChIP assay than that generated by HSF1 antibody (see Fig. 2g). Because of this technical issue, only genome sites abundantly occupied by TRRAP were detected in HeLa cells and the numbers of identified TRRAP binding peaks were limited using ChIP-seq in Fig. 1d. We modified the statement in the results section as below.

“Although the numbers of identified TRRAP binding peaks were limited, they co-occupied 28 sites in heat-shocked cells, including many sites within the gene promoters of major HSPs, co-chaperone (p23), and ubiquitin (UBB) (Fig. 1d-f and Supplementary Fig. 1e, f).” (P6-P7)

We also add the statement in the legend of Fig. 1d as below.

“Because the TRRAP antibody generates a low signal-to-noise ratio in ChIP assay, the limited numbers of TRRAP binding peaks were identified using ChIP-seq.” (P33-P34)

2. In Figure 2A, the authors should describe what the color-coded regions in the model for the HSF1 primary structure are. Although the authors described the DHR domain in previous work (Nakai et al, 1997) what is the significance of this region since that is where Serine 419 appears to lie within? Given that the DHR is conserved in other HSF isoforms outside of HSF1 (see Nakai group previous publications) the lineup in Figure 2e seems to artificially focus on the DHR of HSF1 in different species, but couldn't this region also be functional for other HSF isoforms if the S419 is conserved? Please address this point.

Thank you for careful suggestions. We described the color-coded regions in the model in the legend of Fig. 2a as below.

“DBD, DNA-binding domain; HR, hydrophobic heptad repeat; DHR, downstream of HR-C.” (P34)

In the DHR domain shown in Fig. 2e, the heptad repeats of hydrophobic amino acids (black circles and triangles) are conserved in vertebrate HSF family members (HSF1 to HSF4). In contrast, the S419, which is located in the DHR domain, is evolutionarily conserved in vertebrate HSF1 isoforms, but not in HSF2, HSF3, or HSF4 isoforms. Therefore, we speculate that S419 is functional only for vertebrate HSF1 isoforms.

We modified the statement in the result section as below.

“The S419 is located in the DHR domain and is evolutionarily conserved in vertebrate HSF1 isoforms, but not in HSF2, HSF3, or HSF4 isoforms (Fig. 2e) (Takii et al, *PLoS One* 2017).” (P7)

We also add a statement in the legend of Fig. 2e as below.

“This domain is characterized by the heptad repeats of hydrophobic amino acids (black circles and triangles), which are conserved in vertebrate HSF family members (HSF1 to HSF4).” (P35)

3. In Figure 2C and related primary and supplementary figures showing P-S419 by Western blotting, could the authors point out exactly which band represents the P-S-419 phosphorylated HSF1? It is currently somewhat confusing. Also, in Figure 5a, why is there no heat shock induction of S419 phosphorylation over control levels at 60 minutes, as there is in Figure 2a? This seems inconsistent with the nice time course in Figure 2a.

Thank you for pointing it out. HSF1-S419 phosphorylated bands are detected mainly as two bands, an upper band and a lower band in Fig. 2c, Fig. 2d, Supplementary Fig.2a, and Supplementary Fig.2b. We now indicated the positions of HSF1-S419 phosphorylated bands by blue arrows in these figures, and stated in the legends of these figures that “Blue arrows indicate the positions of HSF1-S419 phosphorylated bands.”.

As we can see, the intensity of the upper band was markedly enhanced during heat shock. However, that of the lower band was enhanced only a little in the same conditions. We stated in the legends of Fig. 2c that “The intensity of the upper band was markedly enhanced during heat shock.”.

The relative intensities of the upper and lower bands are not always constant as the reviewer suggested in Fig. 5a. This may be at least due to differences of anti-HSF1 antibody (the same catalog number, but different lot), which was used to precipitate HSF1 proteins. Nevertheless, we consistently observed enhancement of the intensity of the upper S419-phosphorylated band after heat shock for 60 min in Fig. 5a. We stated again in the legend of Fig. 5a, b and Supplementary Fig. 5a as below.

“Blue arrows indicate the positions of HSF1-S419 phosphorylated bands. The intensity of the upper band was markedly enhanced during heat shock.”.

4. The authors could discuss more about the significance of PLK-1 regulation during the cell cycle. What is the significance of this observation with respect to HSF1 regulation? Could it be related to decreased chromatin access during mitosis as reported by others? Cell cycle regulation of PLK-1 has already been established in the literature at both the protein and mRNA level.

Thank you for constructive comments. PLK1 is a central regulator of cell division that controls a variety of mitotic processes. Therefore, many questions come up about roles of the PLK1-mediated HSF1 phosphorylation in mitotic processes as the reviewer suggested. Because in this manuscript we have no data related to mitotic processes, we hesitate to discuss this point. Instead, we would like to focus on the role of PLK1 in the interphase of the cell cycle. To emphasize this, we showed Supplementary Fig. 5c that PLK1 was not only maximally expressed in the G2/M phase of the cell cycle but also expressed at low levels in the G1 and S phases in the cells we used. The role of PLK1 in HSF1 access to chromatin during mitosis and related questions will be studied in future.

5. The authors accurately state that a previous report showed that PLK-1 phosphorylates HSF1 at Serine 419 in vitro. However, this same reference (Kim et al. 2005, Journal of Biological Chemistry) also presented data to support (1) an interaction between HSF1 and PLK-1 in vivo, (2) that this interaction increases in response to heat shock stress, and (3) that PLK-1 function positively influences HSF1 nuclear accumulation, which could be related to the open chromatin phenomenon beautifully deciphered in the current manuscript. Although in the report by Kiim et al. the mechanisms underlying the role of PLK-1 were not elucidated, the authors could incorporate these previous observations into their manuscript. This would not detract from the novelty of their findings, but rather the prior findings would strongly support the elegant mechanistic dissections that the current authors have done on the role of PLK-1 in the assembly of the HSF1 complex.

Thank you for careful suggestions. We agree with the reviewer, and re-wrote the sentence in the result section as below.

“This result was consistent with previous findings showing that HSF1 interacts with PLK1 in vivo and this interaction increases during heat shock, and PLK1 can phosphorylates S419 in vitro and positively influence HSF1 nuclear accumulation (Kim et al, *J. Biol. Chem.* 2005).” (P13)

6. Page 15, line 269: the term “highly phosphorylated” seems to not adequately describe the observation made. The authors could more accurately describe what was observed in terms of the level of phosphorylation observed.

Thank you for pointing it out. We re-wrote this sentence as below.

“We found that levels of constitutive S419 phosphorylation were higher in various human melanoma cell lines (MeWo, HMV-1, MMAc) than that in immortalized cells (OUMS-36T) (Fig. 7a).” (P15)

7. Page 17, line 305: “cures” should be “cues”

We amended it.

8. Page 15, line 275: The authors note that HSF1 KD is more inhibitory to cell proliferation in melanoma cells compared to other cancer cells. Could the authors speculate on whether this could be due to the levels of TRRAP, TRIM24, TRIM33, PLK-1 mediated HSF1 Ser419 phosphorylation or other aspects of the levels or function of the complex shown in Figure 7H? What do the authors think might underlie this differential sensitivity?

This question is exactly what we would like to know. The different sensitivity to HSF1 KD in melanoma cells and other cancer cells is not due to expression levels of HSF1 (and PLK1) or phosphorylation levels of HSF1-S419 (see blots in melanoma cell lines and HeLa cells in Fig. 7a). We will examine mechanisms explaining for different dependency on HSF1 in future.

9. From a technical perspective, can the authors clarify whether the chromatin IP and proteomics studies were conducted on cells expressing endogenous levels of HSF1 or were the cells over-expressing HSF1? Please clarify in the text of the manuscript.

As we describe at ChIP-MS procedure in method section, this procedure was conducted on cells expressing endogenous levels of HSF1. We modified a statement in result section as below.

“..., we identified proteins that were coprecipitated with endogenous HSF1 at higher levels under heat shock conditions ...” (P6)

10. The authors don't have to address this, but it would be very interesting to investigate how the chromatin-opening complex on HSF1 might change between genes activated by an acute heat shock target and those activated by oncogenic signaling in cancer cells. Additionally, are the TRAPP, etc. proteins important for genes whose expression is repressed by HSF

Thank you for valuable suggestions for future works. We will investigate whether the chromatin-opening complexes on HSF1 change between genes activated by an acute heat shock target and those activated by oncogenic signaling in cancer cells in future. We will also examine

involvement of the chromatin-opening complexes on HSF1 in genes whose expression is repressed by HSF.

**** Reviewer #3 ****

The topic is highly interesting and timely. The question of how sequence-specific transcription factors promote transcriptional activation is still unresolved, despite much work on it. This manuscript provides a novel mechanism at the molecular level that goes a long way towards addressing this question. As the identified phosphorylation mark is upregulated in tumor cells, it constitutes a potential target for anti-cancer therapies. The experiments are thoroughly conducted and the conclusions are well justified. The methodology is described in detail. I do not have any major comments for improvement.

Thank you for favorable and useful comments. We addressed the reviewer's concerns as described below.

Minor comments:

1. Fig. 1e: The authors should spell out in the legend what an MA plot is.

Thank you for pointing it out. We amended the legend of Fig. 1e as below.

“MA (log ratio versus abundance) plot of ChIP-seq binding intensities for HSF1 and TRRAP in control (R_1) and heat shocked (R_2) cells at the common binding peaks for HSF1-HS/TRRAP-HS (28 peaks). The M and A values of each peak were calculated and plotted, where $M = \log_2(R_2/R_1)$ and $A = \log_2(R_1 + R_2)/2$.” (P34)

2. l. 861, l.898, 925: Does n refer to biological or technical replicates?

Yes. We modified the statement at “Statistical analysis” in Method section as below.

“Error bars represent the standard errors of the means for more than three independent experiments (n = number of independent experiments).” (P27)